# Wnt/β-catenin activation by mutually exclusive *FBXW11* and *CTNNB1* hotspot mutations drives salivary basal cell adenoma

Kim Wong [1], Justin A. Bishop[2,12], Ilan Weinreb[3,4,12], Marialetizia Motta[5,12], Martin Del Castillo Velasco-Herrera [1], Emanuele Bellacchio[5], Ingrid Ferreira [1], Louise van der Weyden [1], Jacqueline M. Boccacino [1], Antonella Lauri [5], Giovannina Rotundo[5], Andrea Ciolfi [5], Saamin Cheema [1], Rebeca Olvera-León[1], Victoria Offord[1], Alastair Droop [1], Ian Vermes[1], Michael Allgäuer [6], Martin Hyrcza[7], Elizabeth Anderson[1], Katie Smith[1], Nicolas de Saint Aubain[8], Carolin Mogler [9], Albrecht Stenzinger [6], Mark J. Arends [10], Thomas Brenn[11], Marco Tartaglia [5,13] & David J. Adams [1,13] ✉

Basal cell adenoma (BCA) and basal cell adenocarcinoma (BCAC) of the salivary gland are rare tumours that can be difficult to distinguish from each other and other salivary gland tumour subtypes. Using next-generation sequencing, we identify a recurrent *FBXW11* missense mutation (p.F517S) in BCA that is mutually exclusive with the previously reported *CTNNB1* p.I35T gain-of-function (GoF) mutation with these mutations collectively accounting for 94% of BCAs. In vitro, mutant FBXW11 is characterised by defective binding to β-catenin and higher protein levels within the nucleus. This is consistent with the increased nuclear expression of β-catenin and activation of the Wnt/β-catenin pathway. The genomic profiles of BCAC are distinct from BCA, with hotspot *DICER1* and *HRAS* mutations and putative driver mutations affecting PI3K/AKT and NF-κB signalling pathway genes. These findings have important implications for the diagnosis and treatment of BCA and BCAC, which, despite histopathologic overlap, may be unrelated entities.

Approximately 4–9% of all head and neck neoplasms are salivary gland tumours (SGTs), which include a wide range of benign and malignant entities, that are further classified into distinct histopathological subgroups. Due to their rarity, overlapping presentation, and the difficulty of their diagnosis, salivary gland basal cell adenoma (BCA) and adenocarcinoma (BCAC) are of particular interest amongst these cancers. Cells of the intercalated duct have been proposed to be the primary cells of origin for BCA, a benign tumour[1]. BCAC, its presumed

[1]Experimental Cancer Genetics, Wellcome Sanger Institute, Hinxton, Cambridge, UK. [2]Department of Pathology, University of Texas Southwestern Medical Center, Dallas, Texas, USA. [3]Laboratory Medicine Program, University Health Network, Toronto General Hospital, Toronto, ON, Canada. [4]Department of Pathobiology and Laboratory Medicine, University of Toronto, Toronto, ON, Canada. [5]Molecular Genetics and Functional Genomics, Ospedale Pediatrico Bambino Gesù, IRCCS, Rome, Italy. [6]Institute of Pathology, University Hospital Heidelberg, Heidelberg, Germany. [7]Department of Pathology and Laboratory Medicine, University of Calgary, Arnie Charboneau Cancer Institute, Calgary, AB, Canada. [8]Department of Pathology, Hôpital Universitaire de Bruxelles (HUB), Université Libre de Bruxelles, Brussels, Belgium. [9]School of Medicine and Health, Technical University Munich, Munich, Germany. [10]Edinburgh Pathology, Cancer Research UK Scotland Centre, The University of Edinburgh, Institute of Genetics and Cancer, Edinburgh, UK. [11]Departments of Pathology and Dermatology, University of Michigan, Ann Arbor, Michigan, USA. [12]These authors contributed equally: Justin A. Bishop, Ilan Weinreb, Marialetizia Motta. [13]These authors jointly supervised this work: Marco Tartaglia, David J. Adams ✉e-mail: da1@sanger.ac.uk

malignant counterpart, is thought to arise de novo in most instances, and in rare cases, arises from malignant transformation of BCA[2,3]. Of note, cases of BCA, principally the membranous subtype, and BCAC have been linked to Brooke-Spiegler syndrome[4,5], a rare autosomal dominant condition caused by germline disruption of the tumour suppressor gene *CYLD*. Brook-Spielger is also associated with skin tumors such as cylindromas, spiradenomas, and trichoepitheliomas, 90% involving the head and neck, with histopathological overlaps between these conditions.

The vast majority of salivary gland BCA and BCAC cases arise in the parotid gland[1,6–9]. BCA generally manifests as a slow-growing, well-defined, mobile, painless mass. In contrast, most BCACs are infiltrative and facial nerve involvement occurs in 25% of cases[10]. Local recurrence of BCAC has been reported in up to 50% of patients[11] and occurs within 6-24 months after treatment[12]. Advanced BCAC is occasionally accompanied by cervical lymphadenopathy and distant metastases[13]. The clinical characteristics and natural history of these tumours highlight the need for a precise and prompt diagnosis for effective patient care, with effective surgical intervention being curative and metastatic spread of BCAC clinically challenging to manage. Histopathologically, BCAs are well circumscribed or encapsulated showing tubulotrabecular, cribriform, membranous and/or solid growth patterns. Peripheral palisading of dark cells with luminal paler cells and ducts has been described in BCA, as have vesicular nuclei. The histological patterns of BCAC range widely and includes tubular, membranous, or solid growth patterns, with anaplasia being rare. As with BCA, nuclei are vesicular, while approximately a quarter of BCAC cases are accompanied by perineural and lymphovascular invasion. Diagnosis of BCA and BCAC may be aided by immunohistochemistry (IHC), through the assessment of β-catenin expression and markers of epithelial and myoepithelial differentiation; however, positivity for nuclear β-catenin has been found in both BCA and BCAC[14–16]. In other SGT subtypes, oncogenic, tumour-specific gene fusions have been identified[17–20] and are increasingly used in combination with histology and IHC as a diagnostic aid[21,22]. Genetic studies of BCA have reported a recurrent somatic missense activating mutation (p.I35T) in *CTNNB1*, the gene encoding the β-catenin protein[14,15,23]. Although the *CTNNB1* p.I35T mutation has not been found in BCAC, it has been reported in only a subset of BCA cases (37-80%)[24], and, therefore, is of limited utility as a definitive diagnostic tool. To date, genetic profiling of BCAC cases has revealed mutation of *PIK3CA*, *ATM*, *CYLD* and *NFKBIA*[14,23], but given the rarity of this tumour type and the small number of genes profiled, the roles of these mutations in tumour development is unclear.

In this study, we sequence the tumour and germline exomes and tumour transcriptomes of 32 cases of BCA and 11 cases of BCAC from multiple centres, a sizeable collection given that these tumours represent approximately 1% of all salivary gland neoplasms and arise at a frequency of <1 per 100,000 individuals[25]. We identify and functionally characterise a recurrent somatic missense mutation in *FBXW11*, a gene that encodes a negative regulator of β-catenin, as a driver event in BCA. We show that this recurrent mutation, which is mutually exclusive with the *CTNNB1* p.I35T mutation, impairs proper binding of FBXW11 to β-catenin, indicating the occurrence of a single dysregulated molecular pathway underlying BCA. In BCAC, we identify mutations at known cancer driver hotspots in *HRAS*, *DICER1*, *PIK3CA* and somatic alterations that are predicted to promote tumourigenesis through diverse mechanisms. By examining somatic variants and copy number alterations, germline variants and gene fusions, we provide a comprehensive molecular portrait of these rare and diagnostically challenging head and neck tumours.

## Results
### Case selection and histopathological analysis
We collected formalin-fixed, paraffin-embedded (FFPE) samples from 75 patients, ascertained from six institutions across five countries. After sequencing and quality control, our cohort consisted of 65 cases; 55 cases with tumour and matched normal DNA and 10 tumour-only cases (Supplementary Data 1). Cases underwent a central pathology review by J.A.B. and I.W., who reviewed each case independently, arriving at a consensus diagnosis where their initial classification did not agree. This analysis, together with the genetic information from whole-exome and transcriptome sequencing, such as the presence of fusions diagnostic of other SGT types (described below), resulted in an analysis cohort of 43 cases (Table 1), each with matched normal tissue, including 32 BCAs, 9 BCACs and 2 SGTs with a differential diagnosis of BCAC and epithelial-myoepithelial carcinoma (BCAC/EMC) due to the presence of the oncogenic *HRAS* p.Q61R mutation (Supplementary Fig. 1). Each case was from a different patient. IHC for β-catenin expression was not available for the majority of cases prior to pathology review. Data from transcriptome sequencing was available for all but 1 tumour. The BCAs were from 17 males (mean age of 65.7 years) and 15 females (mean age of 53.9 years), and the BCACs (including 2 BCAC/EMC) were from 6 males (mean age of 55.2 years) and 5 females (mean age of 55.4 years) (Table 1). The majority of the tumours were from the parotid gland. Representative hematoxylin and eosin (HE) stains of a BCA and a BCAC from this study are shown in Supplementary Fig. 2 and Supplementary Fig. 3.

The 20 tumours (12 with matched normal) that were excluded from the final cohort of BCAs and BCACs included diagnostic mimics such as salivary gland pleomorphic adenomas (PAs), adenoid cystic carcinomas (ACCs), an epithelial-myoepithelial carcinoma (EMC), cases with differential diagnoses of PA, myoepithelioma and carcinoma ex-PA (PA/ME/CXPA), and two cases that could not be definitively classified (NOS) (Supplementary Fig. 1). Of note, several PAs were identified to have morphological features indicative of PA with a *HMGA2::WIF1* gene fusion[19], which was subsequently confirmed following the analysis of the transcriptome sequencing data. Other cases had fusions involving *PLAG1*, which is commonly seen in PA/ME/CXPA[20]. Four of the 65 cases had *HRAS* hotspot mutations (three *HRAS* p.Q61R and one p.G13R; Supplementary Fig. 1), which are frequently found in EMCs[26–28], but have also been identified in other SGTs[23,29,30]. Due to this uncertainty, two BCAC cases with *HRAS* p.Q61R were retained for downstream analysis as BCAC/EMC, as described above, and a third, with *HRAS* p.G13R, was classified as a BCAC.

**Table 1 | Summary of salivary gland basal cell adenoma (BCA) and basal cell adenocarcinoma (BCAC) cases**

| Tumour type | DNA samples | RNA samples | Sex | Age, yrs (mean) | Site (Number of tumours) |
|---|---|---|---|---|---|
| BCAC | 11 T/N | 10 T | 5 F<br>6 M | 30–74 (55.2)<br>37–66 (55.4) F<br>30–74 (55.2) M | Parotid gland (7 primary, 1 recurrence)<br>Lung (1 metastasis, 1 salivary gland type carcinoma)<br>Sinonasal (1) |
| BCA | 32 T/N | 32 T | 15 F<br>17 M | 34–83 (60.1)<br>34–71 (53.9) F<br>36–84 (65.7) M | Parotid gland (29)<br>Submandibular gland (2)<br>Cheek (1) |

Number of tumour DNA samples with a matched normal sample, number of RNA samples, age and sex of patients in the BCA and BCAC cohorts used in this study. The BCAC cohort included 2 BCAC with differential diagnosis of epithelial-myoepithelial carcinoma. *T* tumour; *N* normal; *F* female; *M* male. Tumours are primary tumours unless otherwise indicated.

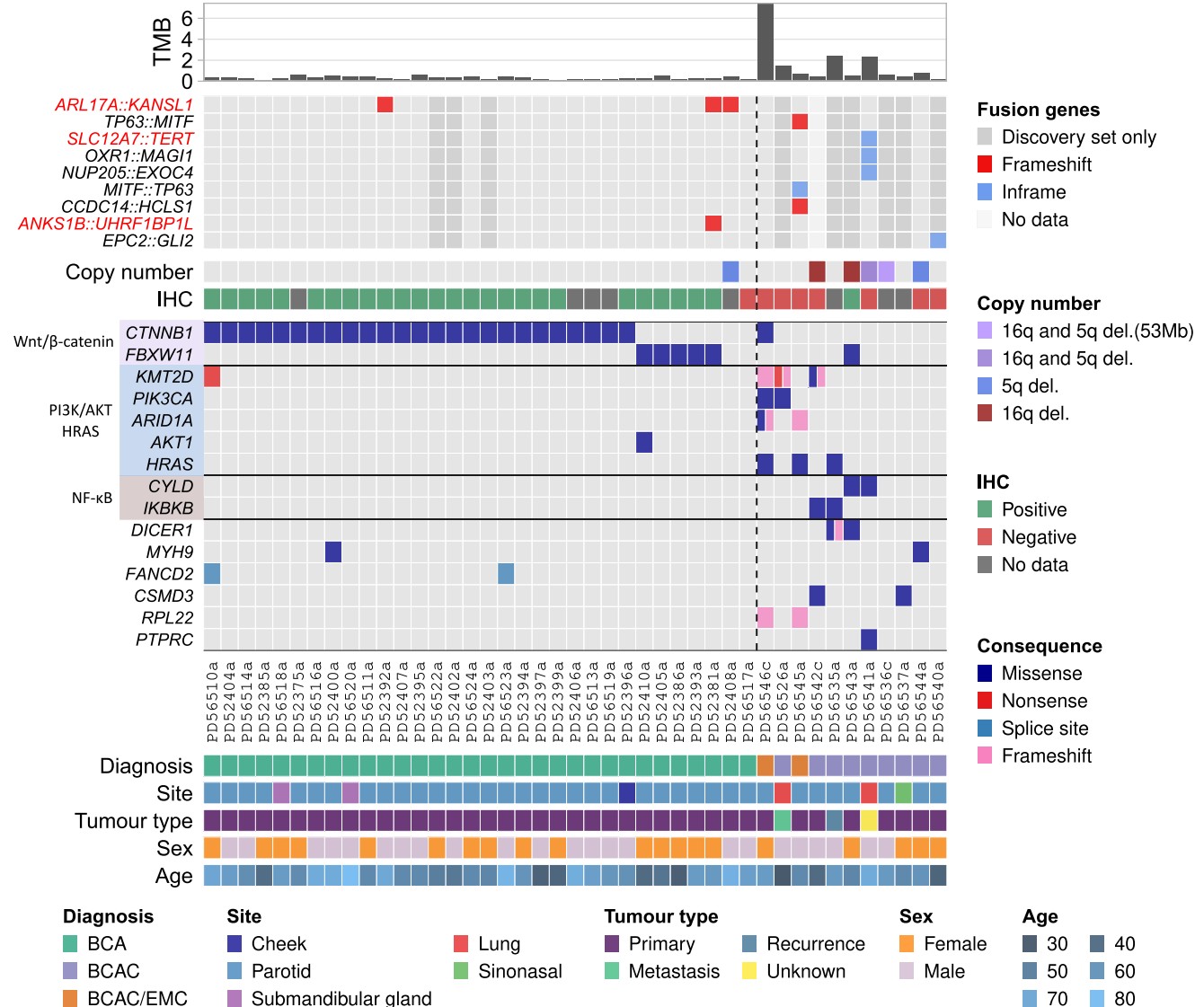

**Fig. 1 | Overview of the salivary gland basal cell adenoma (BCA) and basal cell adenocarcinoma (BCAC) cohorts.** BCA and BCAC cases are shown to the left and right of the dashed line, respectively. Genes shown in the oncoplot are genes known to be mutated in salivary gland tumours and/or are COSMIC Cancer Gene Census (CGC) genes and mutated in at least 1 BCA or BCAC. Also shown is *FBXW11*, which is not a CGC gene, but shown in this study to be significantly mutated in BCA. Genes that were mutated exclusively in the dMMR case, PD56546c, are not shown. Genes in the Wnt/β-catenin, NF-κB, PI3K/AKT or HRAS pathways are grouped by colour. PD56541a is a carcinoma of salivary gland type in the lung, with unknown origin. TMB is the tumour mutation burden in mutations per megabase (Mb). Genes are grouped by pathway. TMB was calculated using mutations found in coding exons and 2 bp upstream and downstream of exon boundaries to account for splice site mutations. Fusion genes in red text are those found in the Trinity Cancer Transcriptome Analysis Toolkit human fusion library (see "Methods"). Samples marked as "Discovery set only" did not pass strict criteria for transcriptome sequencing quality control (see "Methods") and may have a higher false discovery rate as a result. The copy number panel indicates the tumours that had copy number loss of chromosome arms 5q and/or 16q, which was significant in the BCAC cohort. IHC indicates samples that were positive or negative for nuclear β-catenin immunohistochemical staining. BCAC/EMC is differential diagnosis of salivary gland basal cell adenocarcinoma and epithelial-myoepithelial carcinoma. Source data are provided as a Source Data file.

## Overview of somatic alterations in BCA and BCAC

Using whole-exome sequencing (WES) data, we identified somatic single nucleotide variants (SNVs), multinucleotide variants (MNVs) and insertion/deletion variants (indels) (Fig. 1, Supplementary Fig. 4 and Supplementary Data 2). The BCAC cohort ($n = 11$) included 8 primary tumours. Additionally there was one lung metastasis (PD56526a), one carcinoma of salivary gland type in the lung (metastasis of unknown primary vs. lung primary; PD56541a), and one recurrence (PD56535a; BCAC/EMC), all of which had higher mutation rates (1.5-2.4 mutations/Mb) than all but one primary BCAC/EMC (PD56546c; 7.5 mutations/Mb) (Fig. 1, Fig. 2a and Supplementary Table 1). The mutation rate of primary BCAC (0.22–7.5 mutations/Mb) was significantly higher than

BCA (0.12-0.62 mutations/Mb; $n = 32$) (unpaired Wilcoxon rank sum test, $p = 0.0017$), with median mutation rates of 0.56 and 0.32 mutation/Mb, respectively.

Somatic copy number alterations (SCNAs) were infrequent in BCA, with a mean of 1.4% of the genome amplified or deleted (Fig. 2b). In contrast, a mean of 9.7% of the genome of BCACs were altered, with the highest proportion (29%) found in the abovementioned salivary gland type carcinoma of the lung (PD56541a). We identified a chromothripsis-like event (see "Methods") on chromosome 3 of PD56545a, a BCAC/EMC, (Supplementary Fig. 5), however, additional analyses, such as whole-genome sequencing, are required to confirm these events.

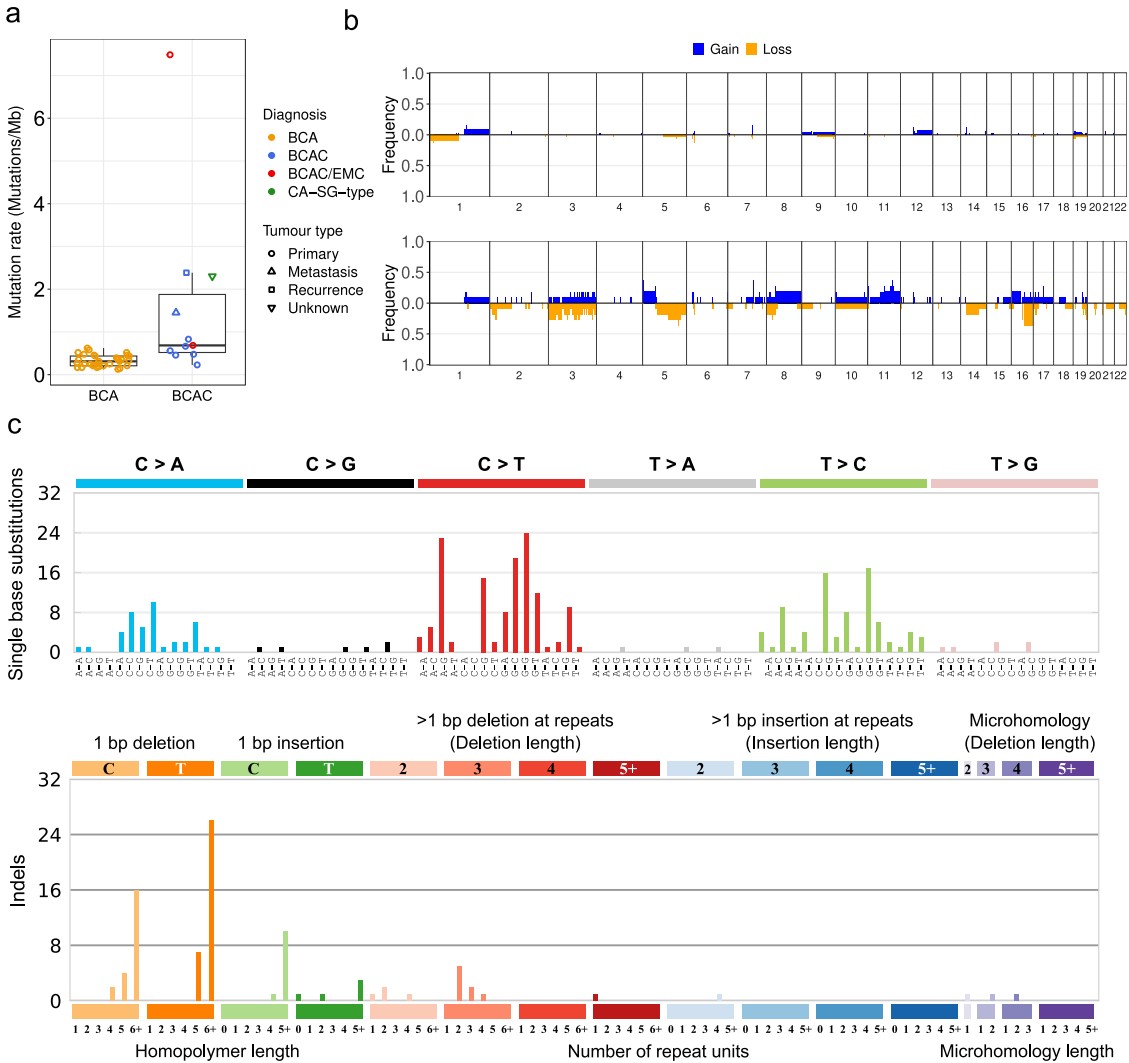

**Fig. 2 | Tumour mutation burden, mutation spectra and recurrent copy number alterations. a** Tumour mutation burden in salivary gland basal cell adenoma (BCA; $n = 32$) and basal cell adenocarcinoma (BCAC; $n = 11$). Boxplot annotations: centre line, median; box limits, upper (75th) and lower (25th) quartiles; whiskers, 1.5x interquartile range; points, tumour mutation burden of samples. **b** Penetrance plot of somatic copy number gains and losses in BCA (top; $n = 32$) and BCAC (lower; $n = 11$). **c** The single base substitution spectrum (top) and indel spectrum (lower) from PD56546c. Show are the number of mutations occurring in each of the 96 possible trinucleotide contexts for single base substitutions and specific repeat contexts for indels.

The BCAC/EMC case (PD56546c) with an elevated mutation rate of 7.5 mutations/Mb (5.7 SNVs/Mb and 1.8 indels/Mb) was attributed to deficiency of mismatch repair (dMMR), as this case had SNV and indel mutational signatures found in tumours with dMMR and microsatellite instability (MSI) (COSMIC signatures SBS44, ID2 and ID7; Fig. 2c and Supplementary Fig. 6) and biallelic inactivation of *MLH1* though somatic disruption of the splice acceptor site of intron 4 and copy number loss of chromosome 3q (Supplementary Fig. 6e). No other COSMIC mutational signatures, other than ubiquitous signatures found in all tumour types (SBS1 and SBS5), were identified in either the BCA or BCAC cohorts.

Unlike other SGTs, such as PA and ACC, recurrent gene fusions were not prevalent in BCA or BCAC (Fig. 1 and Supplementary Data 3). High-confidence fusion genes (see "Methods"), identified through exome-based transcriptome sequencing from tumour samples, were identified in 3 BCAs and 3 BCACs. The only recurrent fusion identified was an inframe *ARL17A::KANSL1* fusion, which was present in 3 BCAs, that resulted from a ~250 kb deletion. Additional experiments are required to validate this fusion and determine its somatic status. The salivary gland type tumour of the lung, PD56541a, harboured 3 gene

fusions, including an inframe *SLC12A7::TERT* fusion. *SLC12A7::TERT* fusions have previously been reported in hepatocellular carcinoma[31] and lung cancer[32].

The high-level comparison of BCA and BCAC presented above alludes to important differences in these tumours. In the following subsections, we further describe and compare the driver genes and pathways in salivary gland BCA and BCAC, and present the results of the functional characterisation of the *CTNNB1* p.I35T (c.104 T > C) mutation found in BCA and a recurrent *FBXW11* mutation identified in this study.

**A recurrent *FBXW11* p.F517S mutation is mutually exclusive with *CTNNB1* p.I35T in BCA**

Previous studies involving a small number of BCAs have identified a recurrent p.I35T activating mutation in *CTNNB1*, which encodes the β-catenin protein[14,15,23]. In our cohort of 32 BCAs, *CTNNB1* was significantly mutated (25/32 or 78% of cases; $q = 0.0$; Benjamini-Hochberg method; see Supplementary Table 2 and Methods), and remarkably all *CTNNB1* mutations were p.I35T (Fig. 1 and Supplementary Data 2). This is in line with previous reports of *CTNNB1* p.I35T occurring in 37-80% of

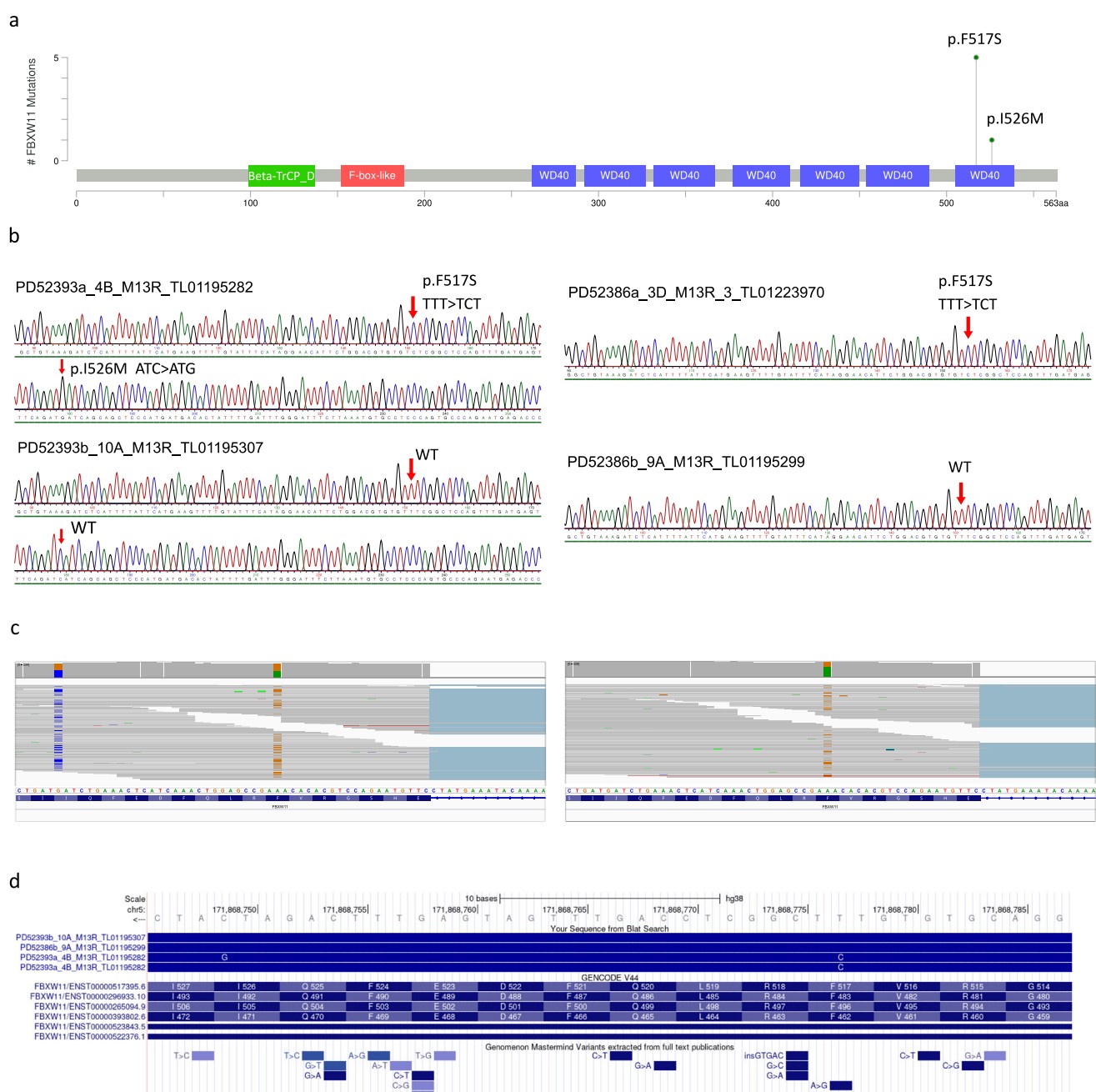

**Fig. 3 | Validation of mutations in *FBXW11*. a** The location and frequency of the p.F517S and p.I526M mutations in FBXW11. The amino acid position is shown on the x-axis, and coloured boxes represent protein domains. Beta-TrCP_D is the D domain of beta-TrCP; also shown are the location of an F-box domain and WD40 repeats. **b** Examples of traces generated from Sanger sequencing of clones derived from PCR amplification of *FBXW11* exon 13 in tumour (top) and matched normal (lower) DNA from PD52393a (left) and PD52386a (right). Tumour PD52393a had a p.F517S (NM_001378974.1:c.1550 T > C) and a p.I526M (c.1578 C > G) mutation. **c** Examples of mutant alleles observed in tumour RNA-seq reads from the patients in (**b**). **d** Visualisation of the alignment of reads from (**b**), patient PD52393, in the UCSC Genome Browser, showing the location of mutant alleles relative to *FBXW11* exon 13. The A > G mutation at p.F517S (relative to transcript ENST00000517395.6) in the Genomenom Mastermind Variants track refers to the report of this mutation in the OKAJIMA cell line.

BCAs[24]. A *CTNNB1* p.D757N (c.2269 C > A) substitution in exon 15, which has not been reported in the COSMIC (v97) database, was found in the dMMR/MSI BCAC/EMC case PD56546c. However, it is unclear whether this mutation is functionally relevant; the majority of *CTNNB1* activating mutations are located in exon 3, which encodes serine/threonine phosphorylation sites that regulate β-catenin degradation[33]. *CDKN1C* and *FGGY* were also statistically significant (*q* = 0.087 for both genes); however, each gene was mutated in only 2/32 BCA cases (Supplementary Table 2).

*FBXW11* was also identified as significantly mutated in BCA (5/32 cases; *q* = 4.0e-05; Benjamini-Hochberg method) and was mutually exclusive with mutation of *CTNNB1* (*q* = 0.012; discrete Benjamini-Hochberg method) (Fig. 1 and Supplementary Table 2). *FBXW11*, located on chromosome 5, encodes a 563 aa protein (Ensembl ID ENSP00000428753.2, encoded by the canonical transcript ENST00000517395.6) that is characterised by an *N*-terminal homodimerization domain, a central F-box region, and 7 tandemly arranged WD40 repeats at the *C*-terminus that constitute a WD repeat (WDR)

domain (Fig. 3a). A recurrent missense mutation in exon 13 of *FBXW11* (NC_000005.10:g.171868777 A > G; NM_001378974.1:c.1550 T > C), resulting in a p.F517S substitution, was identified in all 5 BCA cases with altered *FBXW11* (Fig. 1), with one tumour harbouring an additional *in cis* mutation 28 bp downstream (NM_001378974.1:c.1578 C > G; p.I526M) (Fig. 3b). The *FBWX11* p.F517S mutation was also found in a single high grade, malignant primary BCAC, PD56543a. Importantly, the mutation was also present in the transcriptome sequencing reads from all cases (Fig. 3c). This mutation, which is predicted to be deleterious based on a Sorting Intolerant From Tolerant (SIFT) score of 0 and its location within one of 7 WD40 repeats (Fig. 3a), has not previously been reported in the ClinVar (release date 2023/01/21), dbSNP (v155) or COSMIC (v97) databases, though germline missense variants affecting different residues located in the same domain have been reported, and cause a syndromic neurodevelopmental disorder[34]. A literature search revealed a single report of this mutation in the OKAJIMA gastric cancer cell line[35]. Therefore, we used an orthogonal approach, PCR amplification, shotgun cloning, and Sanger sequencing, to confirm that the *FBXW11* mutation was indeed present in these tumours and absent in the matched normal tissue, indicating that the mutations were of somatic origin (Fig. 3c, Fig. 3d and Supplementary Table 3).

Although there was only a single occurrence of *FBXW11* p.F517S in the BCAC cohort (PD56543a; Fig. 1), the difference in the proportion of BCA and BCAC cases with *FBXW11* p.F517S was not significant ($X^2$ (1, $N = 43$) = 0.16, $p = 0.70$). This may indicate that the p.F517S mutation is not pathognomonic for BCA; however, the possibility that this case represents a malignant transformation of a BCA harbouring the *FBXW11* p.F517S mutation cannot be ruled out. The clinical history of the female patient indicated that she was evaluated at an emergency department (ED) after an 8-month history of having a lump on her cheek, which measured 1.7 cm by computed tomography, and again 6 years later, at which time the lump was 3.4 cm. The patient did not follow up for evaluation with an ear, nose and throat (ENT) specialist. A year later, the patient attended the ED with a 4 cm lump, where it was reported by the patient as having grown larger in recent years. The lump was removed after follow-up with an ENT specialist. The given timeline is consistent with the hypothesis of BCAC arising from malignant transformation of BCA; however, because of the lack of an initial histopathological diagnosis, this cannot be confirmed. Molecular profiling of additional cases of BCAC is required to determine whether the *FBXW11* p.F517S mutation is exclusive to BCA or recurrently mutated in both BCA and BCAC.

In 2 BCA cases, PD56517a and PD52408a, both the recurrent *CTNNB1* and *FBXW11* mutations were absent, as were mutations in other Wnt signalling pathway genes (Supplementary Table 4) and COSMIC Cancer Gene Census genes. *HMCN1* was the only gene recurrently mutated in either of these cases and other BCAs. This gene was mutated in PD565408a and one other BCA with a *FBXW11* p.F517S mutation (PD52386a) (Supplementary Fig. 4).

## Copy number loss of chromosome 5q and 16q is frequent in BCAC

Amplification of oncogenes and deletion of tumour suppressors are known molecular mechanisms of tumourigenesis[36]. SCNAs have been profiled in common salivary tumours[37], with distinct SCNAs found in different subtypes. To our knowledge, SCNAs have been described in one BCAC[38] and one BCA[39], respectively, using comparative genome hybridisation (CGH) and array CGH.

Using WES data, we generated allele-specific copy number (CN) profiles and identified significant recurrent CN alterations in the BCA and BCAC cohorts (Supplementary Table 5). In the BCA cohort, 1 focal deletion and 1 focal amplification were significantly recurrent ($q = 0.022$ and $q = 0.0048$, respectively; Benjamini-Hochberg method; Supplementary Table 5; see "Methods"); however, these regions harbour clusters of paralogous genes, making CN prediction from WES

difficult. A previous study revealed high level amplification of *HMGA2* and/or *HMGA2::WIF1* fusions in CXPA and PA[40]. A ~900 kb SCNA at 12q14.4 was found in case PD565408a, which did not harbour either recurrent *FBXW11* or *CTNNB1* mutation. This low-level amplification, with a total copy number of 5, was predicted to encompass the first 9 of 10 *WIF1* exons and the first 4 of 5 *HMGA2* exons. This case, however, lacked any evidence of *HMGA2* fusion genes from RNA sequencing, and did not have the distinct morphology associated with *HMGA2::WIF1* rearrangements in PA[19]. SCNAs were not found in PD56517a, the only other BCA without a *FBXW11* or *CTNNB1* mutation.

Within our BCAC cohort, we identified 1 significantly recurrent focal deletion encompassing a 18 Mb region on chromosome 2 that included the tumour suppressor genes *DNMT3A* and *ASXL2* (Supplementary Table 5). Sequencing of additional BCAC cases may be required to provide statistical power. Chromosome arms 16q and 5q were significantly deleted ($q = 8.6e-07$ and $q = 0.097$, Benjamini-Hochberg method; see "Methods") in 4/11 and 2/11 samples, respectively (Fig. 1). Of note, 2 BCACs (one lung metastasis and one primary tumour) had biallelic inactivation of *CYLD*, which is located at 16q12.1, through loss of heterozygosity (LOH) and point mutation. Both point mutations were in the ubiquitin carboxyl-terminal hydrolase domain (Supplementary Fig. 7), which is involved in deubiquitination of K63-linked TRAF proteins and negative regulation of NF-κB signalling[41]. *APC* and *FBXW11* are encoded on 5q, but no somatic or germline mutations in these genes were found and no other somatically mutated genes were shared amongst the samples with CN loss of 5q.

## Recurrently mutated genes and pathways in BCAC

While mutations in *PIK3CA* (p.H1047R), *NFKBIA* and deletion of *CYLD* have been identified in BCAC by either targeted panel sequencing or PCR and sequencing of individual genes[14,15,23], whole-exome sequencing in this study has provided further insight into the genes and pathways altered in BCAC. In our cohort, *KMT2D*, *HRAS*, *CYLD*, *IKBKB* and *PIK3CA* were identified as significantly mutated genes ($q < 0.1$; Benjamini-Hochberg method; Supplementary Table 2; see Methods). *RPL22* was significant ($q = 0.028$; Benjamini-Hochberg method) when considering indel variants only. Our cohort of 11 BCACs included 2 cases of BCAC/EMC that each harboured *HRAS* p.Q61R hotspot mutations (Fig. 1). The third case with an *HRAS* mutation (p.Q13R), PD56535a, was diagnosed as recurrent BCAC.

The histone transferase *KMT2D*, also known as *MLL2*, is a tumour suppressor and downstream target of the PI3K/AKT pathway[42] that is frequently mutated in several tumour types[43]. In addition to the dMMR case PD56546c, two other cases harboured 2 *KMT2D* loss-of-function (LoF) mutations each; PD56526a, a lung metastasis with p.R3082Gfs*15 (c.9244_9245del) and p.C778* (c.2334 C > A), and PD56542c, a primary tumour with p.C1408S (c.4222 T > A; located within a PHD-finger and predicted to be deleterious with a SIFT score of 0) and p.F600Cfs*322 (c.1797_1821del; Fig. 1, Supplementary Fig. 7 and Supplementary Data 2), which strongly suggests that biallelic inactivation of *KMT2D* may be a mechanism of tumourigenesis and/or progression in BCAC. Two cases also had hotspot GoF mutations in *PIK3CA*; the helical domain mutation p.E542K (c.1624 G > A) in PD56526a and the kinase domain mutation p.H1047R (c.3140 A > G) in PD56546c (Fig. 1 and Supplementary Fig. 7). A known recurrent *PIK3CA* GoF mutation, p.G118D (c.353 G > A), has been reported in one BCAC[14]. *PIK3CA* encodes the p110α subunit of the phosphoinositide 3-kinase (PI3K) pathway, which plays a role in cell survival and growth, and these hotspot mutations have been reported in a variety of tumour types[44]. Mutations were also observed in *ARID1A*, which can regulate PI3K/AKT pathway activity[45].

In addition to the *FBXW11* p.F517S and *CTNNB1* p.D757N mutations in 2 cases described above, mutations in genes involved in regulation of Wnt/β-catenin and/or NF-κB signalling were found in 4/11 cases (36%) in the BCAC cohort, with mutations in *CYLD* (described above) and *IKBKB*. *CYLD* regulates the NF-κB and Wnt/β-catenin pathways, and

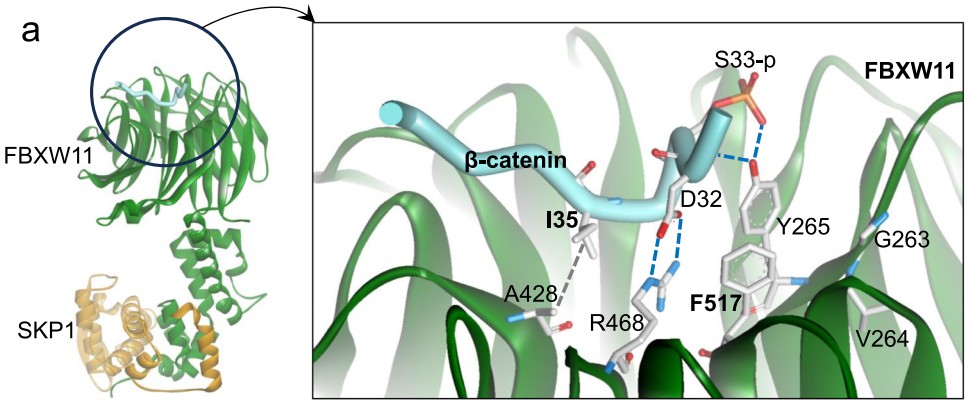

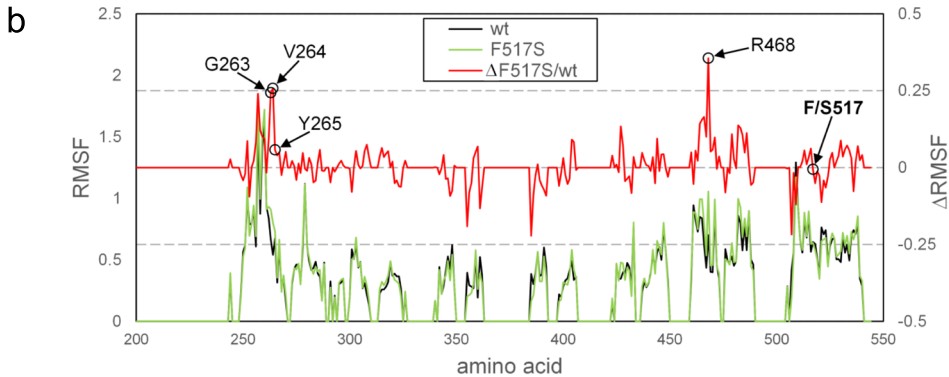

**Fig. 4 | Structural analyses. a** Location of the recurrently mutated sites in β-catenin (I35) and FBXW11 (F517) in the FBXW11-SKP1 complex with a β-catenin peptide phosphorylated on S33 and S37 (PDB ID 6WNX). Relevant residues contributing to the intermolecular binding network stabilising the complex are indicated with their lateral chains. **b** Root mean square fluctuation (RMSF) plot showing the effects of the p.F517S substitution on the conformation of the region of the FBXW11 WDR domain involved in β-catenin binding. The residues at the two highest peaks of RMSF changes are indicated. The F-to-S substitution at position 517 increases the mobility of R468 and a small amino acid stretch adjacent to Y265, which are key residues in the interaction with β-catenin. Source data are provided as a Source Data file.

loss of *CYLD* enhances Wnt/β-catenin signalling[41]. Biallelic inactivation of the NF-κB inhibitor alpha gene, *NFKBIA*, has been found in a BCAC[14], however, we did not observe any non-silent somatic mutations affecting this gene. The inhibitor of nuclear factor kappa B kinase subunit beta gene, *IKBKB*, which encodes the IKKβ protein, was also mutated in 2 cases, with both harbouring missense mutations at codon 171 (p.K171T and p.K171M) within the protein kinase domain (Fig. 1, Supplementary Fig. 7 and Supplementary Data 2). Both variants are represented in the COSMIC database (COSV60598802 and COSV100645808, respectively) and have been previously identified in 3 lymphoid neoplasm samples each. Other substitutions at codon 171 have been shown to constitutively activate IKKβ, which in turn activates the NF-κB signalling pathway[46]. IKKβ has also been shown to phosphorylate β-catenin, which targets the protein for ubiquitination and degradation[47,48].

*DICER1* is a tumour suppressor gene that encodes a ribonuclease involved in the processing of small RNAs, including those that play a role in RNA interference[49]. *DICER1* mutations were identified in 1 primary and 1 recurrent BCAC (Fig. 1, Supplementary Fig. 7 and Supplementary Data 2). The recurrent BCAC, PD56535a, harboured 2 somatic *DICER1* mutations, including the p.E1813D hotspot mutation located in the RNAse IIIb domain[49] and a frameshift deletion at codon 1458 in the region encoding the RNAse IIIa domain. This biallelic hit of *DICER1* is similar to the two-hit pattern of *DICER1* mutations observed in tumours in patients with *DICER1* tumour predisposition syndrome, in which

patients harbour two *DICER* mutations, a germline LoF variant and second mutation affecting the RNAse IIIb domain[50,51]. A second BCAC, PD56543a, had a p.S822T missense mutation located between the dsRNA-binding domain and PAZ domain. Neither patient had a pathogenic germline *DICER* variant.

In summary, we have found a variety of candidate driver genes and pathways in BCAC, whereas the majority (94%) of BCA cases harboured one of 2 mutually exclusive hotspot mutations in genes involving the Wnt/β-catenin signalling pathway. The presence of either mutation in BCA correlated with β-catenin IHC nuclear positivity. There is also a lower tumour mutation burden and few SCNAs in BCA compared to BCAC, in which we found recurrence of 16q and 5q copy number loss. These finding are summarised in Supplementary Fig. 8. These distinctions, along with histopathology, may be useful in developing diagnostic biomarkers.

## Germline and viral risk factors

Little is known about genetic risk factors for developing SGTs. Patients with Brooke-Spiegler syndrome, a rare, inherited condition attributed to germline disruptive variants in the *CYLD* gene, develop tumours primarily on the face, scalp and neck, with cylindromas, spiradenomas and trichoepithelioma as the most common presentation, and rarely, SGTs[52,53]. Although somatic *CYLD* mutations were present in 2 BCAC, we did not identify any pathogenic germline variants in *CYLD* in either the BCA or BCAC cohort.

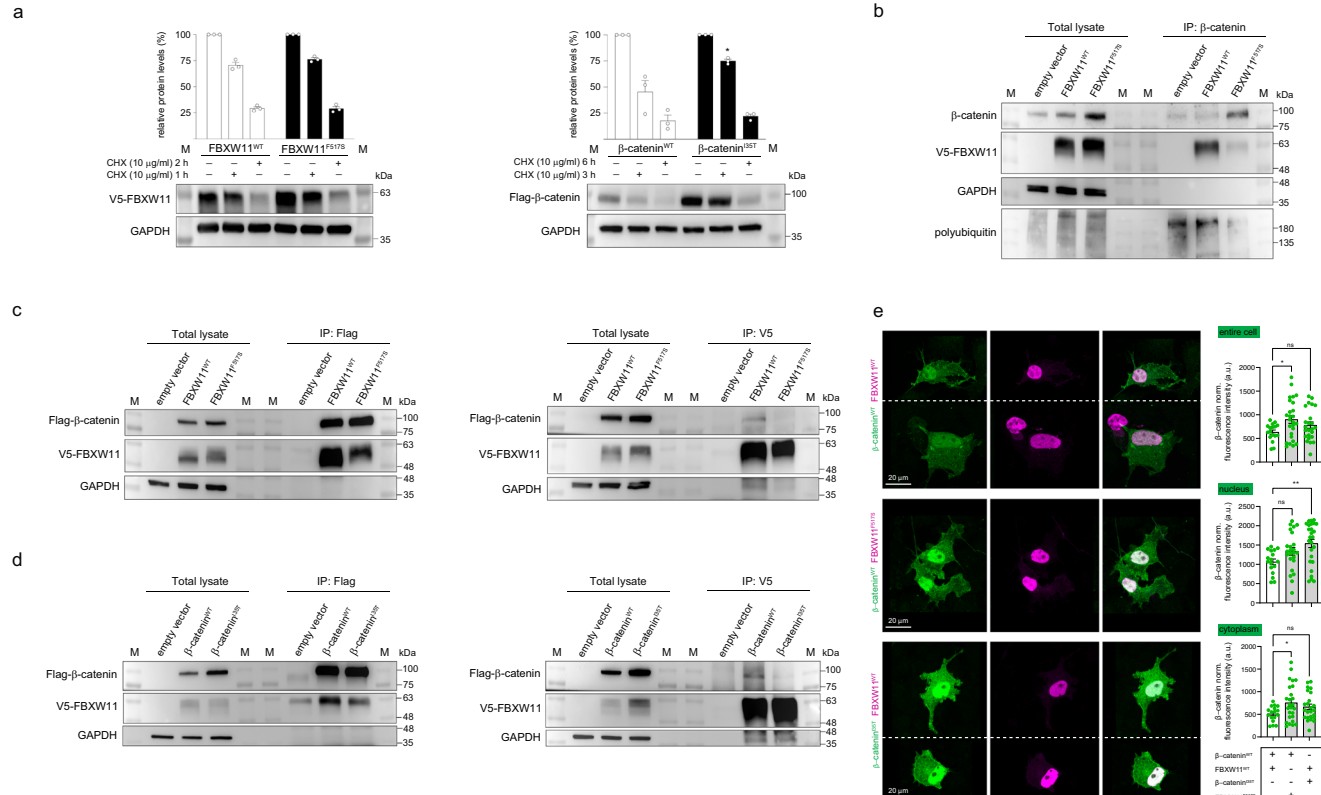

**Fig. 5 | Functional consequences of the BCA-associated *FBXW11* p.F517S and β-catenin p.I35T substitutions. a** FBXW11[F517S] and β-catenin[I35T] are stable proteins. Western blot (WB) and quantitative analyses of V5-tagged FBXW11 and Flag-tagged β-catenin levels in transfected COS-1 cells, basally and after cycloheximide (CHX) treatment. GAPDH was the loading control. M, protein markers. Representative blots shown; data are expressed as mean ± SEM of three biological replicates. Two-way ANOVA followed by Sidak's multiple comparison test was used to test significance (*, *p* = 0.047). **b** FBXW11[F517S] has impaired binding to β-catenin. Lysates from transfected HEK293T cells expressing V5-tagged FBXW11[WT] or FBXW11[F517S] proteins and serum-starved were immunoprecipitated with anti-β-catenin antibody and assayed by WB (two biological replicates). **c** Lysates from transfected HEK293T cells co-expressing FBXW11[WT] or FBXW11[F517S] with Flag-tagged β-catenin and serum-starved were immunoprecipitated with anti-Flag or anti-V5 antibody and assayed by WB (two biological replicates). **d** β-catenin[I35T] has impaired binding ability to FBXW11. Lysates from transfected HEK293T cells co-expressing β-catenin[WT] or β-catenin[I35T] with V5-tagged FBXW11[WT] and serum-starved were immunoprecipitated with anti-Flag or anti-V5 antibody and assayed by WB (two

biological replicates). **e** Relative protein levels and subcellular localisation of serum-starved COS-1 cells co-expressing V5-tagged FBXW11 and Flag-tagged β-catenin revealed by confocal microscopy analysis. Representative images from confocal maximum z-projections of cells co-expressing β-catenin[WT] and FBXW11[WT] (top panels), β-catenin[WT] and FBXW11[F517S] (middle panels), and β-catenin[I35T] and FBXW11[WT] (lower panels). Cells were stained with anti-Flag (green) and anti-V5 (magenta) antibodies. Merged images are shown in the right panels. Scale bar: 20 µm. Bar plots reporting the quantification of the green fluorescent intensity relative to β-catenin expression for the different experimental groups within the entire cell, nucleus or cytoplasm. For each cell, fluorescent intensity is normalised to the cell size (area). Data are expressed as mean ± SEM; *n* = 17 cells (β-catenin[WT] and FBXW11[WT]), *n* = 26 cells (β-catenin[WT] and FBXW11[F517S]), *n* = 24 cells (β-catenin[I35T] and FBXW11[WT]). Brown-Forsythe and Welch one-way ANOVA followed by Dunnett's multiple comparison *post hoc* test was used to test statistical significance (*, *p* = 0.0131; **, *p* = 0.0014; * *p* = 0.0194). Source data are provided as a Source Data file.

Interestingly, some salivary and mammary tumours share morphologic and genetic features[54] and studies have suggested that a previous SGT is a risk factor for developing breast cancer[55–58]. Similarly, a link between germline *BRCA1* and *BRCA2* variants and an increased risk of developing SG tumours has been reported[59]. In our BCAC cohort, *BRCA1* frameshift variants were present in the germlines of patients PD56541 (rs80357906) and PD56542 (rs80357971) (Supplementary Table 6 and Supplementary Fig. 9). Both variants are reported in the ClinVar database (ClinVar Variant ID 17677 and 17667) as pathogenic breast and/or ovarian cancer susceptibility variants that have been reviewed by an expert panel. The tumours from these patients (one primary tumour and one lung metastasis) share somatic loss of 16q (Fig. 1) but no other commonly mutated genes. Two BCA patients and 1 BCAC patient had germline *BRCA2* missense variants of uncertain clinical relevance. Pathogenic germline nonsense mutations in *PMS2* (rs63750451) and *MSH6* (rs267608094) were found in 2 patients (Supplementary Table 6 and Supplementary Fig. 9), however, there was no indication of dMMR in the patients'

tumours, as the tumour mutation burden in these samples was < 1 mutation/Mb.

Human papillomavirus (HPV) has been identified in head and neck squamous cell carcinoma (HNSCC), with 70% of HNSCC of the oropharynx positive for HPV[60]. Consistent with previous findings that HPV is not associated with SGTs[60,61], we did not identify HPV or other viruses associated with BCAC or BCA (see "Methods").

### The *FBXW11* p.517S substitution is predicted to cause local perturbations that affect the β-catenin binding site

Given that the *FBXW11* p.F517S mutation has not been previously reported, we sought to functionally characterise this mutation along with the *CTNNB1* p.I35T mutation. Activation of the Wnt/β-catenin pathway is a known mechanism of tumourigenesis that results from dysregulation of phosphorylation-mediated polyubiquitination and proteolysis of β-catenin, the protein encoded by *CTNNB1*. FBXW11 functions as a substrate adaptor of the SKP1-cullin-F-box (SCF) ubiquitin ligase complex, which catalyses phosphorylation-dependent

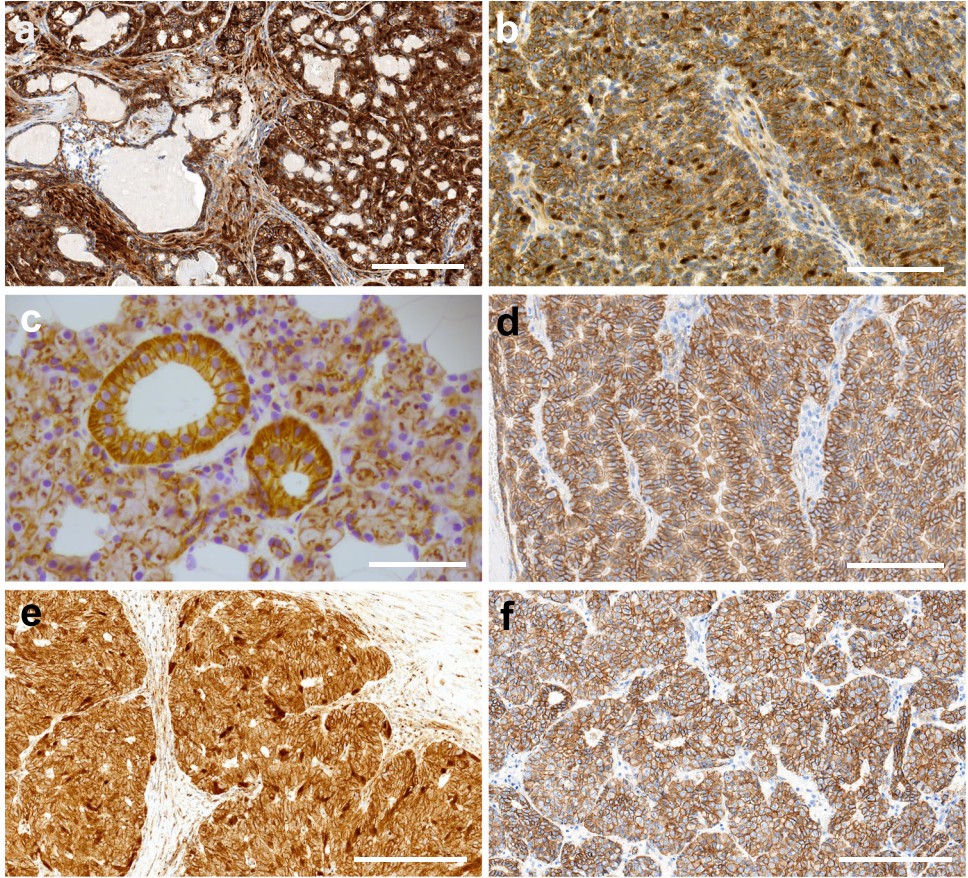

**Fig. 6 | Immunohistochemical patterns of β-catenin in salivary gland basal cell adenoma (BCA) and basal cell adenocarcinoma (BCAC). a** Case PD56510a, a BCA with a *CTNNB1* p.I35T mutation, demonstrating patchy nuclear β-catenin expression in both abluminal basal cells and stromal cells. Scale bar: 200 μm. **b** Case PD52381a, a BCA with a *FBXW11* p.517S mutation, exhibiting patchy nuclear β-catenin expression in a random pattern, along with focal nuclear expression in stromal cells. Scale bar: 100 μm. **c** Normal salivary gland exocrine gland cells from case PD52386a, with membranous β-catenin expression. Scale bar: 50 μm. **d** Case PD56517a, a BCA without a *CTNNB1* or *FBXW11* mutation, exhibiting membranous β-catenin expression. Scale bar: 100 μm. **e** Case PD56543a, a BCAC with a *FBXW11* p.517S mutation, showing patchy, random nuclear β-catenin expression. Scale bar: 200 μm. **f** Case PD56526a, a BCAC showing only membranous β-catenin expression. Scale bar: 200 μm.

ubiquitination of a wide array of substrates, including β-catenin[34,62–64]. Specifically, FBXW11 recognises and binds to phosphorylated β-catenin[62]. The SCF[FBXW11] complex is a component of the β-catenin destruction complex involving AXIN, APC, CK1 and GSK3, which negatively regulates β-catenin levels in the absence of binding of Wnt ligands to membrane receptors[65]. Following Wnt stimulation, β-catenin accumulates in the cytoplasm and translocates to the nucleus where it activates Wnt target genes. Mutations in destruction complex components, such as those involving APC or AXIN, stabilise β-catenin, allowing its translocation to the nucleus[65]. Notably, the *FBXW11* p.F517S mutation is in the last of seven WD40 repeats at the *C*-terminus of FBXW11, which constitute the substrate binding WDR domain (Fig. 3a). Thus, we hypothesised that the *FBXW11* p.F517S mutation may affect binding to β-catenin.

To investigate the functional effects of the p.F517S mutation, we first performed structural inspection of the available crystal structure of the FBXW11-β-catenin complex and multiple short molecular dynamics (MD) simulations, followed by energy minimisations, on wild-type (WT) and F517S structures of FBXW11 (see "Methods"). F517 is in a solvent-exposed region in the WDR domain in the β-catenin binding site, and CTNNB1 D32 and G34 are key sites for the interaction of β-catenin and FBXW11[33,66] (Fig. 4a). F517 constrains a spatially close residue, R468, located in the sixth WD40 repeat, in a position that favours the formation of a salt bridge with D32 in β-catenin. The MD simulations predicted that the replacement of F517 with serine

increases R468 mobility and the adoption of alternative conformations, causing this residue to move away from the β-catenin D32 residue (Fig. 4a).

The proximity of these key residues to F517 prompted us to explore the possibility of local rearrangements/perturbations involving the solvent-exposed region adjacent to the mutated residue with possible impacts on the β-catenin binding site. The MD simulations predicted that the replacement of F517 with serine increases R468 mobility and the adoption conformations alternative to that favouring the binding to the β-catenin D32 residue. In fact, among the residues of the WDR domain (residues 261 to 539), R468 was the residue exhibiting the largest variation in mobility (expressed as root mean square fluctuations, RMSF) associated with the F517S substitution (Fig. 4b). Based on the MD simulations, this change was predicted to disfavour the salt-bridge that contributes to the stabilisation of the FBXW11-β-catenin complex, supporting the hypothesis that the p.F517S substitution reduces the ability of FBXW11 to bind to β-catenin. RMSF analysis also allowed us to identify another region within the first WD40 repeat showing the second highest flexibility change. Notably, this short amino acid stretch includes residues G263 and V264, which are adjacent to the Y265 residue that participates in the β-catenin-FBXW11 intermolecular binding network. These analyses support the idea of local conformational perturbations affecting the β-catenin binding site are caused by the F-to-S substitution at codon 517.

Consistent with the MD simulations, structural analysis of the FBXW11-β-catenin complex showed that the *CTNNB1* p.I35T substitution disrupts the hydrophobic interaction between amino acid residues I35 of *CTNNB1* and A428 of *FBXW11* (Fig. 4a), and could potentially perturb the phosphorylation of adjacent residues S33 and S37. Phosphorylation at these sites, which are part of the β-catenin ubiquitination recognition site, is necessary for the targeting of β-catenin for proteasomal degradation[67,68]. This finding is consistent with β-catenin IHC nuclear positivity in samples harbouring the *CTNNB1* p.I35T mutation observed here and in other studies[14,15,23].

### Impaired binding of mutant FBXW11 to β-catenin leads to reduced polyubiquitination and degradation of β-catenin and its translocation to the nucleus

To validate the functional effects of the recurrent *CTNNB1* and *FBXW11* mutations, we first verified the stability of the mutant proteins. As shown, transient expression experiments in COS-1 cells treated with cycloheximide (CHX) ruled out the occurrence of accelerated degradation for both FBXW11F517S and β-cateninI35T proteins (Fig. 5a). Next, we performed in vitro experiments to confirm that the FBXW11F517S causes a defective FBXW11-β-catenin interaction and subsequent accumulation of β-catenin as a result of its reduced polyubiquitination and degradation. Co-immunoprecipitation (co-IP) assays were performed using lysates collected from overnight-starved (serum-free Dulbecco's Modified Eagle Medium) HEK293T cells transiently expressing V5-tagged FBXW11WT or FBXW11F517S (Fig. 5b), or co-expressing either FBXW11WT or FBXW11F517S together with Flag-tagged β-catenin (Fig. 5c). We observed decreased levels of co-immunoprecipitated β-catenin in cells expressing the FBXW11F517S mutant compared to cells expressing FBXW11WT, indicating a defective formation of the FBXW11F517S-β-catenin complex. This finding was also confirmed in cells overexpressing Flag-tagged β-catenin (Fig. 5c). Consistent with the model in which the decreased ability of FBXW11F517S to bind β-catenin impairs β-catenin

degradation, we observed reduced polyubiquitination and consequent accumulation of endogenous β-catenin in cells expressing the mutant protein compared to the FBXW11WT-expressing controls (Fig. 5b). Similarly, co-IP experiments performed using lysates collected from overnight-starved HEK293T cells transiently co-expressing Flag-tagged WT or mutant β-catenin together with V5-tagged FBXW11 demonstrated a reduced binding of β-cateninI35T to FBXW11 compared with the WT protein (Fig. 5d). To investigate whether the increased levels of β-catenin in cells expressing FBWX11F517S or β-cateninI35T was associated with increased nuclear β-catenin levels, we performed confocal laser scanning microscopy analysis and cell fractioning experiments using COS-1 cells co-expressing each mutant together with its cognate ligand (Fig. 5e and Supplementary Fig. 10). As shown, in both assays, a significant increased level of β-catenin in the nucleus was observed in cells expressing FBXW11F517S as well as in those expressing β-cateninI35T.

Given that in vitro, cells expressing FBXW11517S accumulate β-catenin, we expected to observe nuclear expression of β-catenin in BCA tumours with the *FBXW11* p.F517S mutation. Thus, we performed IHC staining for β-catenin on 21 BCA cases with *CTNNB1* p.I35T, 5 cases with *FBXW11* p.F517S and 1 case with neither mutation (Fig. 1 and Fig. 6). All cases we tested that harboured a mutation in either gene appeared similar, all having little or no membranous positivity with variable mild to moderate cytoplasmic staining and variable strong nuclear positivity, particularly in peripheral palisading dark cells around tumour cell islands. This pattern is consistent with increased accumulation and expression of β-catenin in the BCA tumour cells. Accordingly, analysis of the RNA sequencing data showed relatively higher levels of expression of β-catenin and Wnt/β-catenin targets in tumours with either the *CTNNB1* p.I35T or *FBXW11* p.F517S mutation compared to those without these mutations (Supplementary Fig. 11). In contrast, all but one BCAC case for which IHC for β-catenin was performed (1/8) had membranous positivity and negative nuclear staining (Fig. 1). The exception was the BCAC that harboured the *FBXW11* p.F517S mutation,

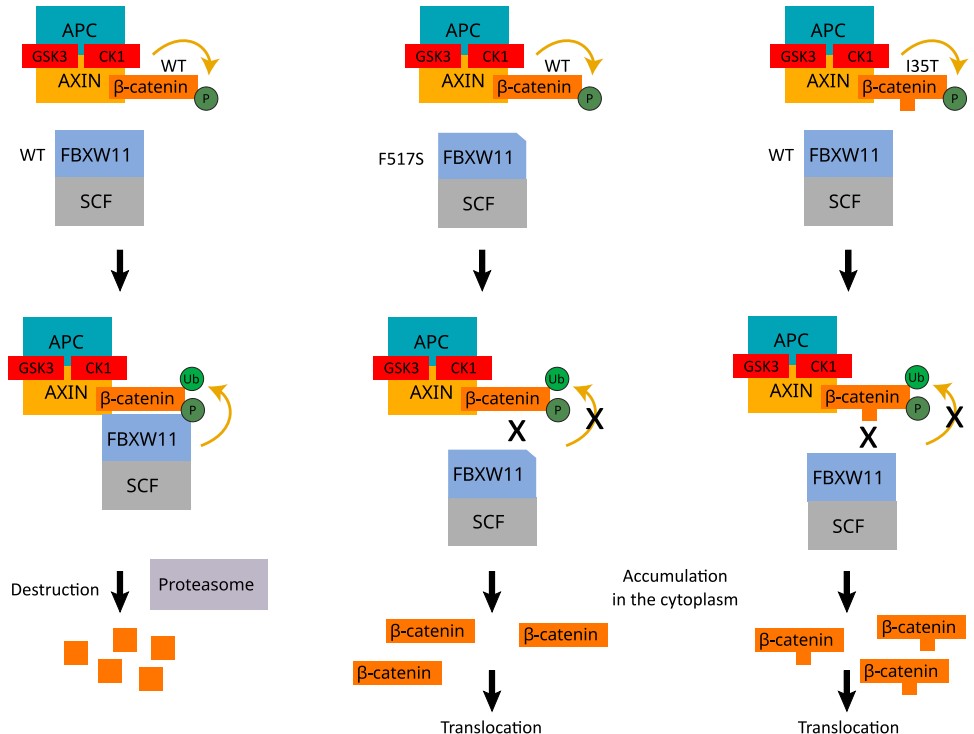

**Fig. 7 | Schematic drawing representing the mechanism of Wnt/β-catenin pathway activation by FBXW11F517S and β-cateninI35T (CTNNB1I35T).** FBXW11 is part of the Skp1-cullin 1-F-box (SCF) complex that regulates β-catenin levels by promoting its ubiquitination and subsequent proteasomal degradation (left). FBXW11 functions as substrate adaptor and is required for binding of

phosphorylated β-catenin to the SCF complex. β-catenin phosphorylation is mediated by the destruction complex, which includes APC, AXIN, GSK3 and CK1. The reduced binding of FBXW11F517S to β-cateninWT (middle) and FBXW11WT to β-cateninI35T (right) inhibits the activity of the SCF complex, leading to accumulation of β-catenin in the cytoplasm and its subsequent translocation to the nucleus.

which demonstrated patchy positive nuclear staining (Fig. 6). Normal salivary gland exocrine gland cells showed no nuclear staining (Fig. 6). A BCA with neither recurrent *CTNNB1* or *FBXW11* mutation was also negative for nuclear staining. Other than *CTNNB1* and *FBXW11*, no other mutations in other Wnt/β-catenin signalling pathway genes (Supplementary Table 4) were detected in the BCA cohort. The discovery of the *FBXW11* p.F517S mutation and evidence that it activates the Wnt/β-catenin pathway therefore explains previous reports of β-catenin nuclear expression in some BCA cases that lacked a *CTNNB1* p.I35T mutation[14,15,23].

A schematic summary of the findings from our validation experiments, IHC studies and RNA sequence analysis is shown in Supplementary Fig. 12. Taken together, our findings strongly suggest that for 30/32 BCA cases (93.7%), activation of the Wnt/β-catenin signalling pathway, either through mutation of *CTNNB1* at p.I35T or *FBXW11* at p.F517S (Fig. 7) is the primary mechanism of tumourigenesis in this tumour entity. Notably, this molecular mechanism does not represent a major driver event in BCAC.

## Discussion

SGTs are rare, clinically heterogeneous and diagnostically challenging, with 39 tumour entities in the World Health Organisation's 5th edition of salivary gland tumour classifications[3]. Here, we focused on BCAC, a rare malignant SGT, and BCA, which is considered its benign counterpart. To date, a limited number of genetic studies performed with small gene panels[14,23] have examined these presentations. By using a multi-omics approach, we provide a comprehensive understanding of the different mechanisms driving BCAC and BCA. We show that a hotspot mutation identified in this study, *FBXW11* p.F571S, and the previously identified *CTNNB1* p.I35T are mutually exclusive in BCA. Our functional characterisation data, IHC studies and examination of RNA expression provides evidence that reduced binding of FBXW11 to β-catenin is the mechanism by which these mutations cause dysregulation of the Wnt/β-catenin signalling pathway. Our study also emphasises the challenge of diagnosing SGTs and the importance of molecular profiling, as a subset of cases originally diagnosed as BCA or BCAC were re-classified as other subtypes after consensus review of the histopathology in conjunction with genetic data.

The tumour suppressor *APC* is part of the destruction complex that regulates β-catenin, and in CRCs, *APC* inactivating mutations are almost always mutually exclusive with *CTNNB1* activating mutations[69]. Although we did not identify *APC* mutations, WES revealed a recurrent somatic driver mutation, *FBXW11* p.F517S (observed in 16% of BCAs), which was mutually exclusive with *CTNNB1* p.I35T (observed in 78% of BCAs). While *APC* mutations stabilise β-catenin by restricting the recruitment of kinases CK1 and GSK3 to the destruction complex[70], we have shown that both the *FBXW11* p.F517S and *CTNNB1* p.I35T mutations directly reduce the binding of FBXW11 to β-catenin, ultimately resulting in subsequent accumulation of β-catenin. This concurred with the observation that tumours with either mutation were positive for β-catenin nuclear staining and relatively higher expression of Wnt/β-catenin gene targets than tumours without these mutations. Notably, in contrast to what has been observed in CRC, mutations in *CTNNB1* and *FBXW11* were limited to a single hotspot position in each gene. While sequencing of additional BCACs is required to confirm the frequency of *FBXW11* p.F517S, if this mutation is exclusive to BCA (and rare cases of BCAC that have arisen from BCA), our finding suggest that genetic testing for *CTNNB1* p.I35T and *FBXW11* p.F517S may be used to confirm nearly 94% of BCA cases helping to distinguish BCA from other mimickers.

While *CTNNB1* mutations have been well characterised in a variety of tumour types, including liver cancer, CRC and endometrial cancer (reviewed here[66]), this is not the case for *CTNNB1* p.I35T. Our structural analysis and MD simulations showed that the p.I35T mutation either disrupts a hydrophobic interaction with FBXW11 at A428 or perturbs

phosphorylation at the adjacent phosphorylation sites. Similarly, the *FBXW11* p.F517S mutation was expected to disrupt the β-catenin binding site at the last WD40 repeat. Interestingly, a recent study identified germline heterozygous variants of *FBXW11* that underlie a clinically variable neurodevelopmental disorder showing developmental delay/intellectual disability, psychiatric features, and eye, digital, and jaw anomalies as major features[34]. Similar to the present finding, these disease-causing variants were found to cluster at the solvent exposed loops of the surface of the WDR domain representing the substrate-binding region of the protein, suggesting a shared functional perturbation exerted by the mutations converging towards a defective binding to substrates. Though only a small number of individuals carrying germline *FBXW11* mutations have been reported thus far, cancer predisposition does not appear a feature associated with the disease[34]. Since the identified mutations were found to affect different WD40 repeats and individual substrates might adopt different orientations when binding to FBXW11 with a variable contribution of the different WD40 repeats in FBXW11 binding to individual ligands, it is possible that the different missense variants might have differential consequences in FBXW11 substrate binding depending on the identity of the substrate.

In line with previous findings, the genetic profiles of BCACs were more varied and complex than BCAs with candidate driver genes *HRAS*, *KMT2D*, *CYLD*, *PIK3CA* and *IKBKB*. The lack of common driver genes and mutations, other than a single BCAC with *FBXW11* p.F517S, suggests that most BCACs arise de novo rather than from malignant transformation of BCA. In this study, *CYLD* mutations were found in only 2 of 11 BCACs and none of the 32 BCAs. A previous study suggested that 36% of BCAs and 29% of BCACs had nonsense mutations in *CYLD*[71], however, because only 8 of the 45 BCAs and BCACs used in the previous study were paired with matched normal tissue[71], the variants found may potentially be germline variants. In addition to *CYLD*, we identified mutations in *IKBKB*; both genes play roles in the Wnt/β-catenin and NF-κB signalling pathways. However, nuclear expression of β-catenin was negative in cases with these mutations, which suggests that the NF-κB pathway, rather than the Wnt/β-catenin pathway, might be activated via functional dysregulation of *CYLD* and *IKBKB* in BCAC. Previous studies have observed nuclear β-catenin expression in subsets of both BCAC and BCA with a wide range of nuclear positivity rates[14,16,72]. However, nearly all BCAs and only 1 of 8 BCAC cases we tested was positive, all of which could be attributed to the presence of either the *CTNNB1* p.I35T or *FBXW11* p.F517S mutation. The inconsistent findings amongst studies strengthens the argument that a combination of tools including histopathology, IHC and genetic profiling can aid diagnosis of SGTs, particularly rare subtypes.

Somatic CN alterations in BCA and BCAC have not been previously profiled using NGS. In this study, WES revealed that 5 of 11 BCACs had loss of one or both of chromosome arms 5q and 16q. These are also frequent events in a subset of breast cancers[73] and numerous studies have attempted to elucidate whether specific tumour suppressor genes are the target of 16q loss[74]. In BCAC, we found somatic biallelic inactivation of *CYLD* was a consequence of 16q loss in 2 of 4 cases, implicating *CYLD* as the target of 16q loss in BCAC.

Our analysis of BCAC has revealed several areas for further investigation. It would be of interest to determine the frequency of dMMR in BCAC, as these patients may benefit from immunotherapy in the metastatic setting. We also identified *KMT2D* as a candidate driver of BCAC and several candidate driver genes in the PIK3K/AKT and NF-κB pathways. Further exploration into the association between breast cancer and BCAC may be of relevance, given that 2 of 11 BCAC patients in this study had LoF *BRCA1* germline variants and previous studies have suggested a higher risk for breast cancer in patients with SGTs. Deconvolution of the roles of these pathways and genes will benefit patients, as treatment can be informed by genetic profiling of tumours.

In summary, we have used whole-exome and transcriptome sequencing to compare the mutational landscape of BCA and BCAC, tumour types that can be difficult to distinguish from each other and other mimickers. We have identified and functionally characterised a recurrent driver mutation *FBXW11* p.F517S, which is mutually exclusive with the Wnt/β-catenin activating mutation *CTNNB1* p.I35T in BCA. The mutant FBXW11 displayed defective β-catenin binding, leading to the stabilisation of β-catenin, thus, serving as an alternative mechanism of Wnt/β-catenin signalling upregulation. We have also shown that BCAC involves several driver genes and pathways, indicating that most BCACs arise de novo and that BCA and BCAC may be unrelated entities. Importantly, our work has highlighted that molecular profiling, along with histopathology and IHC, can be used for more accurate diagnosis of SGTs.

## Methods

### Sample acquisition and nucleic acid isolation
Ethical approval for the use of these samples and associated data was obtained by a local committee at the institution of origin and via Research Governance at the Wellcome Sanger Institute. This study is part of a larger study that has been approved by the National Health Service Health Research Authority (Research Ethics Committee reference 21/PR/1024, IRAS project ID 304621). All patients provided written, informed consent for inclusion of their tissue samples for this study, without compensation, and patient data were anonymised. Cases consisted of formalin-fixed, paraffin-embedded (FFPE) tissues that had been collected as part of routine diagnostic procedures with the patient's consent. Representative HE-stained sections of all FFPE tissue blocks were independently reviewed by two consultant pathologists (I.F. and T.B.) to confirm diagnoses and to identify areas for sampling (tumour and normal, where possible). All tumour and normal tissue samples were obtained as either 1 mm diameter cores or as unstained 10-micron thick tissue sections attached to glass slides (from which the tumour and normal areas were manually macro-dissected). Genomic DNA and RNA was extracted from the tumour samples (with genomic DNA only extracted from normal samples) using the AllPrep DNA/RNA FFPE Kit (Qiagen), according to the manufacturer's instructions.

### Statistics and reproducibility
No statistical method was used to predetermine sample size. Twenty samples were excluded as diagnostic mimics of BCA and BCAC after review by two head and neck pathologists (I.W and J.A.B), as described in the Results section. Samples were also excluded due to sample quality issues such as insufficient sequencing coverage and contamination (see below).

### DNA sequencing, read alignment and quality control
Sequencing libraries were prepared from FFPE-extracted DNA using a NEB Ultra II RNA custom kit on an Agilent Bravo WS automation system. Unique dual index tags were appended, and the samples were amplified by PCR using the KAPA HiFi Kit (KAPA Biosystems) for a minimum of eight cycles. The libraries were quantified using the Accuclear dsDNA Quantitation Kit (Biotium), pooled (8-plex) in an equimolar fashion and hybridised overnight with SureSelect Human All Exon V5 baits (Agilent). The multiplexed samples were paired-end sequenced using the NovaSeq 6000 platform (Illumina) to generate 101 bp reads.

Sequencing reads were aligned to the GRCh38 reference genome, using `BWA-MEM` (0.7.17-r1188)[75] and PCR duplicates from the Binary Alignment Map (BAM) file were marked using the `samtools` (v1.14)[76] `markdup` function with parameters `-mode s -S -include-fails`. Matched tumour-normal sample concordance, as well as cross-individual contamination, was assessed using `Conpair` (v0.2)[77]. After excluding samples with quality issues such as having less than 80% of

the bait capture regions with a minimum of 20X coverage, matched tumour-normal genotype concordance < 60%, or cross-individual contamination >5% and selecting one sample per tumour (where applicable), there were 48 cases diagnosed as BCA (including 7 cases without matched normal DNA available) and 17 samples diagnosed as BCAC (including 3 cases without matched normal DNA available). Given that histopathological diagnosis of salivary gland tumours can be challenging, the diagnoses for samples passing quality control were then independently re-reviewed by two specialist head and neck pathologists (J.A.B and I.W.). Slide images were reviewed, along with genetic data, which included fusion genes information associated with diagnostic mimics (for example the *NFIB*::*MYB* fusion in ACC and the *HMGA2*::*WIF1* fusion in PA) and somatic variants in genes previously described in BCA or BCAC, including *CTNNB1*, *PIK3CA* and *HRAS*. In cases where the diagnosis differed after independent review, a consensus diagnosis was derived after a joint review. This detailed case review resulted in a cohort of 11 BCAC and 32 BCA, each with matched normal DNA.

### Exome-capture transcriptome sequencing, read alignment and quality control
FFPE-extracted RNA samples were reverse-transcribed and sequencing libraries prepared using the NEBNext Ultra II Directional RNA Library Prep kit (New England Biolabs) according to the manufacturer's instructions. Unique dual index tags were appended, and the samples were amplified by PCR using the KAPA HiFi HotStart ReadyMix PCR Kit (Roche) for a minimum of 16 cycles. Libraries were quantified using the Accuclear dsDNA Quantitation Kit (Biotium), pooled (8-plex) in an equimolar fashion and hybridised overnight with the SureSelect Human All Exon V5 baits (Agilent). The multiplexed samples were paired-end sequenced using the NovaSeq 6000 platform (Illumina) to generate 101 bp reads. Reads were aligned using `STAR` (v2.5.0c15)[78] against the GRCh38 human reference genome using Ensembl release v103 gene annotations. Expression levels were assessed by counting reads using `HTseq` (v0.7.2)[79] with the appropriate stranded parameter and subsequently transformed into transcripts per million (TPM) values. Data quality was assessed by running `RNA-SeqQC 2`[80] and assessing the total number of counts obtained per sample. The criteria to select samples for a high-quality analysis cohort are provided in the Supplementary Methods. A total of 29 BCA and 4 BCAC RNA sequencing datasets passed these criteria for fusion gene analysis. For consensus review of cases (described above), all cases for which sequencing data was available were used for fusion discovery (32/32 BCA and 10/11 BCAC). We refer to this as the discovery set of samples.

### Identification and annotation of somatic variants
Somatic point mutations were identified using `cgpCaVEMan` (v1.15.2)[81]. The parameters used and input files are described in the Supplementary Methods. Variant flagging and annotation were not performed initially. Instead, adjacent, *in cis* SNVs called with `cgpCaVEMan` were evaluated using `SmartPhase` (v1.2.1)[82] and `casmsmartphase` (v0.1.8; https://github.com/cancerit/CASM-Smart-Phase) to identify MNVs. Variants were then flagged using the `cgpcavemanpostprocessing` (v1.10)[81] `cgpFlagCaVEMan.pl` utility using the 'WXS' mode for exomes. The parameters and flagging rules used are described in the Supplementary Methods.

Indels on the autosomes and chromosomes X and Y were identified using `cgpPindel` (v.3.10.0)[83]. A simple repeats file for GRCh38, generated using the UCSC Table Browser, and a list of regions to exclude due excessive high depth of coverage, were used as inputs. Soft flag `FF017` was used, but variants were not hard filtered based on this flag. Additional details, including a description of parameters, flags and input files, are available in the Supplementary Methods.

The Ensembl release v103 Variant Effect Predictor (`VEP`)[84] was used to predict the consequences of SNVs, MNVs and indels on

proteins. Because some genes have multiple transcripts, the canonical transcript, as defined in Ensembl v103, was used to determine the variant consequence. `VEP` was also used to add custom annotations from the COSMIC (v97)[85], ClinVar (update 20230121)[86], gnomAD (v3.1.2)[87] and dbSNP (v155)[88] databases.

Common SNPs, defined as variants with a minor allele frequency of at least 0.01 in the total population in the gnomAD database (v3.1.2) or the 1000 Genomes Phase 3 dataset (as indicated in dbSNP v155), were excluded as germline variants. `cgpPindel` calls were further refined by retaining variants with VAF ≥ 0.1, multinucleotide variants of length 3 bp or less. Variants were retained where both the reference allele and alternative allele were ≤25 bp in length or more but both alleles were not >10 bp, otherwise the variants were retained only if the tumour VAF > 0.25 and sequencing coverage at the variant site was ≥20 in both the tumour and matched normal. Finally, variants within 100 bp of exome targeted regions and not flagged by `cgpCaVEMan` or `cgpPindel` filters were taken forward for further analysis.

Manual inspection of read alignments at recurrently mutated sites in *CTNNB1* and *FBXW11* was performed to minimise false negative calls at these sites due to the presence of contaminating tumour cells in the matched normal sample and/or low variant allele frequency that may result from low tumour purity.

The TMB was calculated using the total SNVs, MNVs and indels in exons and splice sites of canonical transcripts. The exome target region was 48.225157 Mb when including exons and a 2 bp flank to account for splice sites. The unpaired Wilcoxon ranks sum test was used to compare the TMBs of BCA and primary BCAC tumours. Continuity correction was used for the normal approximation of the *p*-value.

## Somatic copy number alterations

SCNAs were identified using `ASCAT` (v3.1.2)[89] in all autosomes and chromosome X. The required hg38 reference files for processing WES data (loci, allele, GC correction and replication timing correction files) were downloaded from https://github.com/VanLoo-lab/ascat/tree/master/ReferenceFiles/WES (`git` commit ID 29f2fad). Cases with a solution with a goodness-of-fit score < 0.9 were considered noisy and excluded from further analysis.

To find significant recurrent amplifications and deletions, the outputs from `ASCAT` were used to generate input files for `GISTIC2` (v2.0.23)[90], which requires segment coordinates, the number of markers and a log2-scaled copy number. For each segment obtained from segmentation from `ASCAT` analysis, the normalised depth log ratio was extracted from the `ASCAT` output files, and the number of assessed loci within each segment was used as the number of markers. Broad and focal copy number events were defined as events involving at least half a chromosome arm and less than half a chromosome arm, respectively. The residual *q*-value threshold for significant events for both broad and focal amplifications and deletions was set at 0.10. If a warning of low statistical power was given in the output, the result was not used. To remove artefactual recurrent SCNAs, we further removed `GISTIC2` focal events ≤ 100 kb in size and those with an overlap of more than 40% with regions known to be problematic for sequencing and/or read mapping, as defined by the Genome in a Bottle Consortium benchmark union set of all difficult regions (v3.3; https://ftp-trace.ncbi.nlm.nih.gov/ReferenceSamples/giab/release/genome-stratifications/v3.3/GRCh38@all/Union/GRCh38_alldifficultregions.bed.gz). Finally, we excluded a focal or broad event if less than 75% of samples predicted to have a particular SCNA in `GISTIC2` did not concur with the original `ASCAT` results.

## Prediction of chromothripsis events

We applied the definition and scoring method outlined in Vorinina et al.[91] for the identification of putative chromothripsis events in tumour samples, which is based on assessing the number of CN state switches within a 50 Mb region. The segmentation results from `ASCAT`, described above, were used as input. Predicted chromothripsis events were classified as high-confidence (10 or more CN state switches in 50 Mb), intermediate-confidence (8 or 9 CN switches in 50 Mb) or low-confidence (6 or 7 CN state switches in 50 Mb). Chromothripsis events were further classified as canonical (2 or 3 different CN states) or non-canonical (>3 CN states). For chromosome 21, which is ~47 Mb, the number of required CN state switches was scaled accordingly and rounded to the nearest whole number. For example, a scaling factor of 0.934 (46.709/50 Mb) was applied to the definitions of high-confidence (9 or more CN state switches), intermediate-confidence (7 or 8 CN state switches), and low-confidence (6 CN state switches) regions of chromothripsis.

## Identification of significantly mutated genes

To identify driver genes, we used `dNdscv` (v0.1.0; `git` commit ID 64f8443)[92], which detects genes under positive selection in cancer. The reference gene annotations and covariates files used to run `dNdScv` were provided in the `dNdScv` package (RefCDS_human_GRCh38_GencodeV18_recommended.rda and covariates_hg19_hg38_epigenome_pcawg.rda, respectively). The outputs include *p*-values that were adjusted using the Benjamini-Hochberg method, and the *q*-value threshold was set at 0.10. A gene was selected if any of the *q*-values for substitutions and indels combined, substitutions only, missense mutations, nonsense mutations or indels only were below the *q*-value threshold.

`OnodriveFML` (v 2.4.0)[93] was also used to identify candidate driver genes. `OncodriveFML` determines whether the average functional impact score of somatic mutations on an element (in this case, a coding gene) is significantly higher than expected, `OncodriveFML` was run with the following parameters: `genome build, hg38; signature method, none; statistic method, amean; sampling, 100,000; sampling max, 1,000,000; sampling chunk, 100; sampling min obs, 10; include indels, true; indel method, max; indel max consecutive, 7`. Functional impact scores for the genome were downloaded from Combined Annotation Dependent Depletion (CADD; v1.7) website (https://cadd.gs.washington.edu; https://krishna.gs.washington.edu/download/CADD/v1.7/GRCh38/whole_genome_SNVs.tsv.gz). A regions file consisting of 100 bp padded bait set regions was used. A gene was considered significantly mutated if it was mutated in at least 2 samples and had *q* < 0.1. For both `dNdScv` and `OncodriveFML`, mutations in significant genes were verified by visual inspection of read alignments, and genes with artefactual variants were excluded.

## Mutational signature analysis

To identify somatic mutational signatures for single base substitutions, doublet base substitutions and indels, somatic mutations were analysed using `SigProfilerExtractor` (v1.1.21)[94] and `SigProfiler-Assignment` (v0.0.30)[95]. Results from signature decomposition and assignment were taken forward if the cosine similarity between the de novo extracted signatures and signatures reconstructed from assigned COSMIC signatures (v3.3) was .≥ 0.9. Similarly, on a per-sample basis, signature analysis was considered reliable if the cosine similarity between the original mutational spectrum and reconstructed spectrum was .≥ 0.9 and there were greater than 100 mutations per sample, per analysis type (SBS, DBS and ID)

## Mutual exclusivity and co-occurrence of mutated genes

`DISCOVER` (r_v0.9.4)[96] was used to test for co-occurring and mutually exclusive mutation of genes within each tumour subtype independently and within a cohort that consisted of both tumour types. A binary alteration matrix containing genes with non-silent somatic mutations was used for analysis. For both the co-occurrence and mutual exclusivity tests, the `DBH` parameter selected for the estimation

of the false discovery rate (FDR), which implements a discrete Benjamini-Hochberg procedure that is described in Canisius et al. [96]. Genes used for the analysis were mutated in 2 or more samples within the cohort. A FDR threshold of 0.1 was applied.

## Analysis of whole exome germline variants

Following the Genome Analysis Toolkit (GATK) Best Practices workflow[97], PCR duplicates from aligned WES reads in normal sample BAMs were marked using the Picard Tools (v2.25.4; https://broadinstitute.github.io/picard/) MarkDuplicates function and short variant discovery was performed using GATK (v4.2.6.1)[98]. Additional details are provided in the Supplementary Methods. For both SNVs and indels, only variants within 100 bp of a bait set region were retained. Functional consequences were annotated using VEP, as described above for the annotation of somatic variants. To identify variants present within known germline cancer predisposition genes, we looked for variants present within the set of genes used by England's National Health Service (NHS) Cancer National Genomic Test directory (v7.2, June-2023; https://www.england.nhs.uk/wp-content/uploads/2018/08/Cancer-national-genomic-test-directory-version-7.2-June-2023.xlsx) to diagnose cancer predisposition in patients. Only variants with a moderate or high impact on the encoded protein, as predicted by VEP, were reported.

## Fusion gene analysis

Paired-end exome-capture transcriptome sequencing data was used to identify fusion transcripts in tumour samples. Fusion identification was performed using STAR-Fusion (v1.10.1)[99], with STAR (v2.78a) aligner and the Trinity Cancer Transcriptome Analysis Toolkit (CTAT) genome library StarFv1.10 for GRCh38 using GENCODE v37 (Ensembl v103) gene annotations. We used FusionInspector (v2.6.0) included with the STAR-Fusion package to assess the coding effect of the fusions, annotate their presence in the CTAT library and validate them in silico using the STAR-Fusion parameters --FusionInspector validate --examine_coding_effect --denovo_reconstructFusion. For the purpose of aiding case review of the salivary gland tumours collected, in the discovery set of samples we minimally filtered the fusion gene predictions, removing only fusions with less than 5 junction-spanning reads, in order to identity known fusions or putative fusions involving either HMGA2, MYB, NFIB, PLAG1 or WIF1. To identify high-confidence fusions, we searched for fusions in the high-quality cohort (described above). A set of high-confidence fusion predictions obtained by removing those with junction reads < 5; fusions without a coding effect prediction (frameshift or inframe fusion transcript); fusions previously reported in normal tissues in the Genotype-Tissue Expression (GTEx) project database; and fusions involving genes described as neighbours. Lists of recurrent fusions in salivary gland tumours and frequently rearranged genes were obtain from Freiberger et al. [100]. and Bubola et al. [101]. The collation of results, filtering, summaries per cohort and plotting was performed using custom scripts.

## Gene expression analysis

The RNA sequencing data from BCA and BCAC tumours that passed quality control for fusion analysis were also used for gene expression analysis. Additionally, tumours that had been reclassified after consensus review were used as comparators. These samples were subject to the same data quality assessment as the BCA and BCAC samples. Gene expression z-scores were calculated from the TMP values and complete linkage hierarchical clustering of the z-scores using Euclidean distance was performed on Wnt/β-catenin target genes using the R package ComplexHeatmap (v2.14.0)[102].

## Pathogen and micro-organism identification

We used Kraken2 (v2.1.2)[103] to search for evidence of pathogens in the tumour samples. Briefly, unfiltered paired-end exome-capture RNA-seq FASTQ files were provided as input to Kraken2, which was run with a 16GB-capped reference database containing RefSeq sequences from archaea, bacteria, viruses, plasmids, humans, UniVec_Core, protozoa, fungi, and plants (https://benlangmead.github.io/aws-indexes/k2; March 2023). A confidence score threshold of 0.1 was used. Next, we calculated the proportion of minimisers from the Kraken2 reference database found in each sample for each taxon, defined as the number of distinct minimisers out of the total clade level minimisers. To assess the significance of the results, we used a binomial distribution, approximated from a normal distribution, to calculate the probability of finding $m$ minimisers, out of the total $M$ minimisers available for a taxon, in $n$ reads of a sample, considering the total $N$ reads evaluated by Kraken2 in that sample. We used the Benjamini-Hochberg method to correct for multiple comparisons. Results with adjusted $p$-value < 0.05 were considered statistically significant.

## Shotgun cloning and Sanger sequencing

Confirmation of the FBXW11 mutation was performed for 5 tumour samples (PD52386a, PD52405a, PD52410a, PD56543a, PD52393a) and their respective matched normal sample, where DNA stock was available (PD52386b, PD52405b, PD52393b). Validation was not performed for BCA case PD52381a due to the exhaustion of DNA stocks. The region surrounding the FBXW11 p.F517S (g.5:171868777 A > G) variant was amplified using the primers FBXW11_F CAAAGGGAAGGTGAATT-CAATCA and FBXW11_R TGCCCAGTTTCTCATTGTGA in AccuStart II Gel-Track PCR SuperMix with 35 cycles using DNA extracted from FFPE cores as described above. After PCR clean-up, fragments were cloned using the TOPO™ TA Cloning™ Kit and shotgun cloned into Stellar™ Competent Cells (TaKaRa). Individual colonies were picked into 96 well plate culture blocks, prepped for plasmid DNA extraction, and capillary sequenced with M13F and M13R primers. Reads were analysed by manual inspection and by alignment to the human reference genome and to exon 13 of FBXW11.

## Structural analysis

The crystal structure of FBXW11 (Protein Data Bank ID: 6WNX) was used to investigate the functional effect of the p.F517S substitution (Uniprot isoform E5RGC1). Wild-type FBXW11 and a model of the p.F517S protein mutant (including the water molecules present in the crystal structure) were subjected to cycles of energy-minimisation, short molecular dynamics (MD) simulations (1 ns), and re-minimisation with HyperChem (v8.0; Hypercube, Inc., Gainesville, FL). Computations were performed considering all residues within a selection sphere (20 Å from the C-alpha atom of residue 517). The Amber94 force field was used employing switched cutoffs (inner and outer radius were respectively 10 and 14 Å) and constant temperature (309.15 K). One hundred independent MD minimisation cycles were made to produce as many conformers for both wild type FBXW11 and the F517S mutant. RMSF were calculated with VMD (v1.9.3)[104].

## In vitro functional analyses

A complete list of reagents is provided in the Supplementary Methods. Abbreviations are as follows: Dulbecco's modified Eagle's medium (DMEM); Foetal bovine serum (FBS); cycloheximide (CHX).

Constructs. The entire coding sequence of FBXW11 was cloned into the pcDNA6.2/V5-HisA eukaryotic expression vector (Invitrogen). The human β-catenin pcDNA3 plasmid was purchased from Addgene. The constructs expressing the mutant FBXW11[F517S] and β-catenin[I35T] protein were generated by PCR-based site directed mutagenesis using the QuiKChange II Site-Directed Mutagenesis Kit (Agilent), as previously described[105]. All constructs were bidirectionally Sanger-sequenced for their entire open reading frame (ABI BigDye terminator Sequencing Kit v3.1) (Applied Biosystems, Foster City, CA), using a SeqStudio Genetic Analyser (Applied Biosystems).

Cell cultures, transfections and inhibitor treatment. COS-1 and HEK293T cell lines (American Type Culture Collection, Manassas, VA) were cultured in DMEM medium supplemented with 10% heat-inactivated FBS, and antibiotics (37 °C, humidified atmosphere containing 5% CO$_2$). Subconfluent COS-1 and HEK293T cells were transfected using the PEI transfection reagent according to the manufacturer's instructions. COS-1 cells were treated with CHX (10 μg/ml) to assess protein stability. Serum-free DMEM was utilized to starve cells.

Cell lysates and co-immunoprecipitation assays. After treatment with CHX, COS-1 cells were lysed in radio-immune precipitation assay (RIPA) buffer, pH 8.0, complemented with protease and phosphatase inhibitor cocktails. Lysates were kept on ice for 30 min and then centrifuged at 16,000 × $g$ for 30 min at 4 °C. For FBXW11/β-catenin co-immunoprecipitation (co-IP) assays, HEK293T cells were serum-starved and lysed in IP buffer containing 25 mM Tris-HCl (pH 7.4), 1% Triton X-100, 2 mM EDTA (pH 8.0), 150 mM NaCl, supplemented with protease and phosphatase inhibitors. Samples were centrifuged at 10,000 × $g$ (20 min, 4 °C). Supernatants were collected, and their protein concentration was determined by Bradford assay, using bovine serum albumin (BSA) as a standard. In co-IP experiments, equal amounts of total proteins were immunoprecipitated using an anti-β-catenin or anti-V5 or anti-Flag antibody cross-linked to Protein G Sepharose beads (2 h, 4 °C). The beads were recovered by centrifugation and washed six times with IP buffer. Finally, the immunoprecipitated proteins were eluted with sample buffer by incubating at 95 °C (10 min) and stored at -20 °C until immunoblotting analysis.

Immunoblotting assays were performed as previously reported[105]. In brief, cell lysates were resolved by sodium dodecyl sulphate (SDS)-polyacrylamide gel electrophoresis. Proteins were transferred to a nitrocellulose membrane using the Trans-Blot Turbo transfer system. Blots were blocked with 5% non-fat milk powder in PBS containing 0.1% Tween-20 for 1 h and incubated with specific antibodies overnight. Primary and secondary antibodies were diluted in blocking solution. Immunoreactive proteins were detected by an enhanced chemiluminescence (ECL) detection kit, according to the manufacturer's instructions. Densitometric analysis of protein bands was performed using NineAlliance UVITEC software (UVITEC, Cambridge, UK).

Confocal laser scanning microscopy. COS-1 cells (15 × 10$^3$) were seeded on glass coverslips, transfected with the various constructs for 24 h, fixed with 4% paraformaldehyde for 30 min at 4 °C, and permeabilized with 0.5% Triton X-100 for 10 min at room temperature. For double immunostaining, cells were incubated with a mouse monoclonal primary antibody (anti-Flag) overnight at room temperature, rinsed twice with PBS and incubated with a cross-adsorbed secondary goat anti-mouse IgG1 antibody conjugated with Alexa Fluor 488 (A-21121) (Thermo Fisher Scientific) for 1 h at room temperature. Cells were then rinsed three times with PBS, incubated 1 h with a mouse monoclonal primary antibody (anti-V5), rinsed twice with PBS and incubated 1 h with a cross-adsorbed secondary goat anti-mouse IgG2a antibody conjugated with Alexa Fluor 568 (A-21134) (Thermo Fisher Scientific). Finally, nuclei were stained with Hoechst 33342 solution and glass coverslips were mounted on the microscope slides by using PBS-glycerol buffer. Confocal analysis was performed using a Leica Stellaris 5 confocal microscope (Leica Microsystems) equipped with LAS X software v.4.5, using an HC PL APO CS2 ×63 objective, 512 × 512 format at 600 Hz and z-step of 0.3 μm, using 488 nm and 568 nm laser excitation for β-catenin and FBXW11 immunostaining, respectively. ROI-based analysis was performed using the *Fiji* software to measure total cell and nuclear area, and raw integrated density (fluorescence) within cells. For each cell, fluorescence intensity was normalised by cell or nucleus' area. Data were assessed for normal distribution, and statistical significance of the observed distributions was one-way ANOVA test. Image processing used Illustrator 2022 (Adobe Systems Incorporated, San Jose, CA).

Cell fractionation. ProteoExtract Subcellular Proteome Extraction Kit (Calbiochem) was used for subcellular extraction of proteins. Briefly, COS-1 cells (1 × 10$^6$) were seeded in 100 mm petri dishes, transfected with the various constructs for 24 h and serum-starved overnight. Cells were washed twice with wash buffer and incubated with extraction buffer I for 10 min at 4 °C on an orbital shaker. After removing the cytosolic and membrane/organelle fractions, the remaining cellular material was incubated with extraction buffer III containing benzonase nuclease for 10 min at 4 °C, and the resulting supernatant containing the nuclear proteins was recovered. One tenth of each nuclear fraction was subjected to SDS-PAGE followed by western blotting. To ensure no cross-contamination between fractions, the same membrane was probed with antibodies HSP90 (cytoplasmic marker) or YY1 (nuclear marker).

## Immunohistochemistry for β-catenin
β-catenin immunohistochemistry was performed on a Leica BOND III™ stainer (Leica Biosystems) using the validated 'IHC Protocol F' for human tissue, according to the manufacturer's instructions. The primary antibody was a mouse monoclonal anti-human β-catenin antibody (Clone beta-catenin 1, M3539, Dako, Agilent) used at 1:100 dilution.

## Antibodies
Primary antibodies were used as follows: mouse anti-V5 (Invitrogen, cat. no. R96025, clone SV5-Pk1, lot 2735895) was used at 1:2,500 for western blot (WB; overnight at 4 °C) and 1:50 for immunofluorescence (IF; 1 h at room temperature). Mouse anti-GAPDH (Santa Cruz, cat. no. sc-32233, clone 6C5, lot I0319) was used at 1:1,000 for WB (1 h at room temperature). Mouse anti-Flag (Sigma, cat. no. F3165, clone M2, lot SLCP4941) was used at 1:1,000 for WB (overnight at 4 °C), and mouse anti-Flag (Sigma, cat. no. F1804, clone M2, lot SLCD6338) was used at 1:300 for IF (overnight at 4 °C). Mouse anti-β-catenin (Abcam, cat. no. AB22656, clone 12F7, lot 1013722-1) was used at 1:2,500 for WB (overnight at 4 °C). Mouse anti-polyubiquitin (Enzo Life Sciences, cat. no. BML-PW8805-0500, clone FK1, lot 11011759) was used at 1:1,000 for WB (overnight at 4 °C). Rabbit anti-HSP90 (Cell Signalling Technology, cat. no. 4874S, lot 6) and rabbit anti-YY1 (Cell Signalling Technology, cat. no. 46395S, clone D5D9Z, lot 1) were each used at 1:1,000 for WB (overnight at 4 °C).

## Reporting summary
Further information on research design is available in the Nature Portfolio Reporting Summary linked to this article.

## Data availability
Sequencing data are available from the European Genome-Phenome Archive (EGA) under dataset accessions EGAD00001015365 (DNA) and EGAD00001015366 (RNA). Please refer to the Sanger Institute data sharing policy at https://www.sanger.ac.uk/about/research-policies/open-access-science/. These data are freely available to researchers who agree to the EGA guidelines for respecting patient anonymity and the associated data management rules to ensure the data is securely stored, which aligns with the patient consent. Source data are provided with this paper. The processed data generated in this study, such as variant calls, are provided in the Supplementary Information and Source Data files. Source data are provided with this paper.

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

## Acknowledgements

This work was supported by a Medical Research Council (MRC) programme grant to D.J.A. (MR/V000292/1) and Ministero della Salute (5 × 1000_2024) and an AIRC (IG 28768) grants to M.T. This research was funded in whole, or in part, by Wellcome Trust Grant 206194. The authors thank the CASM IT and Pathogens and Microbes teams, Wellcome Sanger Institute, for their assistance in the development of analysis pipelines. We also wish to thank the patients and their families and Dr. Nithila Joseph for laboratory assistance.

## Author contributions

K.W., D.J.A., and M.T. wrote the paper with assistance from all of the authors. K.W., M.D.C.V-H., J.M.B., S.C., V.O., I.V., and A.D. performed analysis of the next-generation sequencing data. J.A.B., I.W., I.F., M.H., M.J.A., N.d.S.A, C.M., A.S., and T.B. performed histopathological and immunohistochemical analysis. R.O-L performed PCR and shotgun cloning for validation of the *FBWX11* mutation. L.v.d.W., E.A., and K.S. managed the sample collections and the generation of the sequence data and were responsible for ensuring ethical compliance along with the pathologists who supplied the cases. M.M., E.B., A.L., G.R., A.C., and M.T. performed the FBXW11 modelling analyses, molecular dynamics simulations, and functional studies.

## Competing interests

The authors have no conflicts to declare.
