## [Transparent Peer Review file · Nature Communications]

Wnt/ β -catenin activation by mutually exclusive FBXW11 and CTNNB1 hotspot mutations drives salivary basal cell adenoma

Corresponding Author: Dr David Adams

Version 0:

Reviewer comments:

Reviewer #1

(Remarks to the Author)

The authors present a robust multi-institutional study on molecular profile of BCA and BCAC of salivary glands. The main finding is newly invented missense mutation in 94% of BCA. It is well designed, written and argued study.

Authors may still expand how 6% of tumors without this findings is characterized. How pre-operative practical use will be applicable? Any targeted treatment?

Are two cases of pulmonary tumors real metastases or primary salivary-like lung tumors?

Reference 10 is confusing: is it book or journal? Please correct.

The paper would benefit from a mind map to show all findings in one figure and clinical potential.

Reviewer #2

(Remarks to the Author)

The manuscript under review entitled "Activation of Wnt/ β -catenin Signalling by mutually exclusive FBXW11 and CTNNB1 hotspot mutations drives salivary gland basal cell adenoma" presents a comprehensive genetic profiling of two types of salivary gland tumors, BCA (benign) and BCAC (malignant carcinoma). Utilizing whole-exome and transcriptome sequencing, the study examines somatic variants, copy number alterations, germline variants, and gene fusions. Among the key findings are the identification of a novel mutational event involving elevated Wnt/ β -catenin Signalling and the observation of the FBXW11 p.F517S mutation, both of which are significant contributions to the field. However, the manuscript suffers from several shortcomings that impact its clarity.

Strengths:

1. Identification of Novel Mutational Events in BCA: The discovery of the FBXW11 mutation and its association with elevated β -catenin levels is a noteworthy finding. This adds valuable information to the understanding of genetic alterations in BCA salivary gland tumors.
2. Insight into Tumorigenesis in BCAC: The study's use of high-throughput sequencing technologies has generated high-quality genetic data, which is particularly important given the rarity of these tumor types. Moreover, the study documented the genetic changes in both the benign and malignant forms of these rare salivary gland tumors, providing new insights into the tumorigenic processes in these tumours.

General Comment:

The manuscript lacks a clear thematic focus. It frequently jumps between descriptions of gene mutations, associated pathways, and different cancer groups, making it difficult to follow. A more structured approach, perhaps separating the description of genetic findings from mechanistic analyses, would enhance readability.

Specific points:

1. While the investigation of β -catenin accumulation by FBXW11 pF517S is interesting, the study falls short by not

conducting further experiments to demonstrate the effect of this mutation on β -catenin subcellular localization and its transactivation activity.

2. The mutual elusive nature of CTNNB1 at p.I35T and FBXW11 at p.F517S in BCA samples is observed but not further explored. The MD simulations showed that the CTNNB1 p.I35T substitution disrupts the hydrophobic interaction between amino acid residues I35 of CTNNB1 and A428 of FBXW11 (Figure 4a). Disruption of the interaction between CTNNB1 p.I35T and FBXW11 could be experimentally validated.

3. The study presented the MD simulation analysis of FBXW11 at p.F517S and β -catenin. To demonstrate F517S affects the FBXW11 region involved in binding β -catenin, providing protein-ligand interaction stability throughout the simulation by RMSD (root mean square deviation) as well as RMSF (root mean square fluctuation) data of protein which depicts the flexibility for the wildtype and mutant protein when binding to β -catenin would greatly strengthen the conclusion. Additionally, providing the protein-ligand contacts throughout trajectory would improve clarity of the data.

4. To strength the conclusion that 30/32 BCA cases (93.7%) arose from accumulation and activation of the Wnt/ β -catenin signalling pathway, through mutation of CTNNB1 at p.I35T or FBXW11 at p.F517S, the levels of β -catenin and FBXW11 in cancer tissues harboring the mutations of CTNNB1 at p.I35T or FBXW11 at p.F517S should be examined accompanying the IHC staining of β -catenin.

In summary, while the manuscript presents significant genetic findings in the context of salivary gland tumors, its value is diminished by several issues listed. Future revisions should focus on enhancing the narrative structure, and incorporating functional experiments to elucidate the role of identified mutations in tumorigenesis.

Reviewer #3

(Remarks to the Author)

Wong et al. present "Activation of Wnt/ β -catenin signaling by mutually exclusive FBXW11 and CTNNB1 hotspot mutations drives salivary gland basal cell adenoma". It is a very nice study of the somatic genomic alterations in two extremely rare cancer entities of the salivary gland. Overall, the study is very well done, following-up on the most interesting new finding (the mutation in FBXW11) with computational and in vitro studies that support its potential oncogenic role.

I only have a few minor comments:

1 - There's a sentence to close one of the results sections that reads as follows:

"Taken together, we can conclude that 30/32 BCA cases (93.7%) arose from activation of the Wnt/ β -catenin signalling pathway, either through mutation of CTNNB1 at p.I35T or FBXW11 at p.F517S (Figure 1). Notably, this molecular mechanism does not represent a major driver event in BCAC."

I believe that this might be a bit too strong. While it is true that all the evidence suggests a very important role for these two mutations, there is no experimental demonstration that the BCA cases "arose from activation" of the Wnt pathway. I know that this is extremely hard to actually prove, but given the growing number of papers showing driver mutations in otherwise healthy cells, I believe that the expression "arose from" is not yet deserved. Maybe something like: "30/32 cases had alterations in the Wnt pathway, suggesting a very important role of this biological process in the tumorigenesis of BCA"?

2 - There's a variety of tools to identify cancer driver genes. Why did the authors only use dn/ds? This can lead to missing important genes.

3 - There is a sentence in the discussion: In this study, CYLD mutations were found in only 2 of 11 BCAs and none of the 32 BCACs.

I believe that here BCA / BCAC is switched

Version 1:

Reviewer comments:

Reviewer #2

(Remarks to the Author)

Following the revisions made by the authors, I am pleased to provide my review of the revised manuscript.

- The overall flow and structure of the manuscript have been markedly improved.
- Additional experimental data have been provided to strengthen the conclusion that mutations of CTNNB1 at p.I35T and FBXW11 at p.F517S in BCA samples affect the interaction between the two proteins leading to nuclear translocation and accumulation of β -catenin

The manuscript now addresses the major concerns raised in the initial review.

Reviewer #3

(Remarks to the Author)

The reviewers have addressed all my comments

Reviewer #4

(Remarks to the Author)

The authors present a significant multi-institutionally collected, well validated, series of patients with rare basal cell adenomas (BCA) and basal cell adenocarcinomas (BCAC) originating from the salivary glands. Using whole-exome and whole-transcriptome sequencing as well as in vitro experiments they find a novel missense mutation of FBXW11 and show that it is mutually exclusive with CTNNB1 GoF mutations in BCAs. They also showed that the novel FBXW11 mutation led to accumulation of beta-catenin in the nucleus. They further show that the genomic profile of BCAC is distinct from BCA with mainly harboring known and more frequent hotspot mutations and therefore suggesting de-novo development of BCAC rather than malignant transformation in most cases.

Overall, it is a well designed and conducted study on very rare salivary gland tumors/carcinomas resulting in a good manuscript. Translation of the findings into clinical practice may help in the differential diagnostics of salivary gland tumors. Sequencing of FBXW11 and CTNNB1 as well as IHC of beta-catenin may help to verify the diagnosis of BCA after validation of these findings.

The queries raised by reviewer 1 were sufficiently addressed:

The genomic status of the two BCA patients without FBXW11/CTNNB1 was investigated and described within the manuscript, potential practical implications of the findings were discussed, information on the two cases within the lung was added, and a mindmap summarizing the findings for differential diagnosis between BCA and BCAC was added.

Giving these improvements, I recommended acceptance for publication of the present manuscript.

Reviewer comments and author responses

Reviewer #1

Remarks to the Author:

The authors present a robust multi-institutional study on molecular profile of BCA and BCAC of salivary glands. The main finding is newly invented missense mutation in 94% of BCA. It is a well designed, written and argued study.

Authors may still expand how 6% of tumors without this findings is characterized.

We thank Reviewer 1 for their comments in support of our study.

The recurrent *CTNNB1* p.L35T and *FBXW11* p.F517S mutations were absent from 2 of the 32 cases of BCA (PD56517a and PD52408a). Of note, for PD56517a, where we had IHC for β -catenin expression, nuclear staining was negative* (*Note: Figure 1 has now been corrected to indicate negative nuclear staining, as stated in the main text). Importantly, previous studies have reported that the majority, but not all, BCA are positive for nuclear β -catenin expression (see references Vickie *et al.*, 2016, Karahawa *et al.*, 2011). Notably, all cases in this study have been reviewed by two specialist head and neck pathologists (J.A.B and I.W; see Methods), and the diagnosis of BCA confirmed by each pathologist independently. We are thus confident that they are BCAs. Neither PD56517a nor PD52408a showed any distinctive histopathological features. For reference, the membranous IHC staining for β -catenin in PD56517a is shown in Figure 6d.

To further explore PD56517a and PD52408a, we searched for mutations in a list of genes in the Wnt signalling pathway (Supplementary Table 7). We did not find mutations in these genes. Further, there were no commonly mutated genes in PD56517a and PD52408a, but we note that *HMCN1* was mutated in PD52408a and one other BCA (Supplementary Figure 4). As we have not identified any candidate driver genes in these cases, this small subset of BCA cases will rely on histopathology, patient history and clinical presentation for their accurate diagnosis. We have added the following text to the manuscript (line 289):

“In 2 BCA cases, PD56517a and PD52408a, both the recurrent *CTNNB1* and *FBXW11* mutations were absent, as were mutations in other Wnt signalling pathway genes (Supplementary Table 7) and COSMIC Cancer Gene Census genes. *HMCN1* was the only gene recurrently mutated in either of these two cases and other BCAs. This gene was mutated in PD565408a and one other BCA with an *FBXW11* p.F517S mutation (PD52386a) (Supplementary Figure 4).”

In the section discussing somatic copy number alterations, we have added this text:

“A previous study revealed high level amplification of *HMGA2* and/or *HMGA2::WIF1* fusions in CXPA and PA⁴⁰. A ~900kb SCNA at 12q14.4 was found in case PD565408a, which did not harbour either recurrent *FBXW11* or *CTNNB1* mutation. This low-level amplification, with a total copy number of 5, was predicted to encompass the first 9 of 10 *WIF1* exons and the first 4 of 5 *HMGA2* exons. This case, however, lacked any evidence of *HMGA2* fusion genes from RNA sequencing, and did not have the distinct morphology associated with *HMGA2::WIF1* rearrangements in PA¹⁹. SCNAs were not found in PD56517a, the only other BCA without a *FBXW11* and *CTNNB1* mutation.”

How pre-operative practical use will be applicable? Any targeted treatment?

The results presented here can be used to ensure an accurate diagnosis after removal of the tumour. This is important when making decisions about patient treatment, screening and follow-up, and to exclude other possible mimics, such as an adnexal skin tumour that has spread to the salivary gland. Other salivary gland tumour types, for example, have tumour-specific fusions that aid in diagnosis of those tumour types. These points have been made in the Introduction (line 106):

“Diagnosis of BCA and BCAC may be aided by immunohistochemistry (IHC), through the assessment of β -catenin expression and markers of epithelial and myoepithelial differentiation; however, positivity for nuclear β -catenin has been found in both BCA and BCAC^{14–16}. In other SGT subtypes, oncogenic, tumour-specific gene fusions have been identified^{17–20}, and are increasingly used in combination with histology and IHC as a diagnostic aid^{21,22}. Genetic studies of BCA have reported a recurrent somatic missense activating mutation (p.I35T) in *CTNNB1*, the gene encoding the β -catenin protein^{14,15,23}. Although the *CTNNB1* p.I35T mutation has not been

found in BCAC, it has been reported in only a subset of BCA cases (37-80%)²⁴, and, therefore, is of limited utility as a definitive diagnostic tool.“

and in the Discussion (line 653):

“While sequencing of additional BCACs is required to confirm the frequency of *FBXW11* p.F517S, if this mutation is exclusive to BCA (and rare cases of BCAC that have arisen from BCA), our finding suggest that genetic testing for *CTNNB1* p.I35T and *FBXW11* p.F517S may be used to confirm nearly 94% of BCA cases, helping to distinguish BCA from other mimickers.“

Additionally, the mind map added to the Supplementary Materials (Supplementary Figure 8), based on the reviewer’s suggestion, summarises the similarities and differences between BCA and BCAC, which can inform the differential diagnosis of these tumours. The ability to correctly diagnose benign BCA or malignant BCAC is essential, since BCACs are more aggressively treated than BCA. Additional comments relating to the mind map are provided below in the response to the reviewer’s suggestion to include a mind map of this study.

Are two cases of pulmonary tumors real metastases or primary salivary-like lung tumors?

We have reviewed the clinical details from these two cases. For case PD56526a, the patient had a basal cell adenocarcinoma of the parotid gland in 2013 treated with surgery and radiation. In 2019 (at a different hospital) the patient presented with local recurrence and pulmonary metastases to the right lower lobe. The hospital received a resection of the right upper lobe (in 2019) containing two foci of cancer that were compatible with metastases of basal cell adenocarcinoma of the parotid. Therefore, there is every reason to believe that this is a metastasis and not a primary salivary gland type tumour of the bronchus.

There is no history of a primary tumour in the clinic details for patient PD56541, therefore, we cannot be completely certain whether this is a metastasis or primary salivary-like lung tumour. Consequently, we have changed the classification to “carcinoma of salivary type in lung (metastasis of unknown primary or lung primary)”.

In the manuscript, the following changes were made:

1. Section “Overview of somatic alterations in BCA and BCAC” (line 181):

“The BCAC cohort included 8 primary tumours, one lung metastasis (PD56526a), one carcinoma of salivary gland type in the lung (metastasis of unknown primary vs. lung primary; PD56541a) and one recurrence (PD56535a; BCAC/EMC), all of which had higher mutation rates (1.5-2.4 mutations/Mb) than all but one primary BCAC/EMC (PD56546c; 7.5 mutations/Mb) (Figure 1, Figure 2a and Supplementary Table 3).”

2. We have also corrected the references to sample PD56541a throughout the text (see tracked changes in the manuscript), Table 1, Figure 2, Supplementary Table 1, Supplementary Table 3, Supplementary Table 9, Supplementary Figure 1, Supplementary Figure 4 and Supplementary Figure 5.

Reference 10 is confusing: is it book or journal? Please correct.

Thank you for finding this error. The correct reference is:

“Ellis, Gary L. & Auclair, Paul L. *Atlas of Tumor Pathology. Tumors of the Salivary Glands, 3rd Series, Fascicle 17.* (Armed Forces Institute of Pathology, 1996).”

We have corrected the reference in the manuscript.

The paper would benefit from a mind map to show all findings in one figure and clinical potential.

Based on the reviewer’s suggestion, we have created a mind map of our findings based on the analysis of sequencing data and immunohistochemistry for β -catenin. This mind map helps emphasise the fact that BCA and BCAC are genetically distinct, despite having an overlap in their histopathological features. The ability to differentiate between benign BCA and malignant BCAC is essential for patient care. BCACs are more aggressively treated, with surgical excision with a wide margin and, in some cases, postoperative radiotherapy. The findings presented here

suggest that genetic testing for the *CTNNB1* and *FBXW11* recurrent mutations can corroborate 94% of BCAs and, therefore, be used for differential diagnosis between BCA and BCAC in most cases. Inclusion of the novel *FBXW11* mutation in a targeted sequencing panel, such as SalvGlandDx, would facilitate in differentiating BCA from BCAC and other mimickers in a clinical setting. We have included the mind map in the manuscript as Supplementary Figure 8, and we have included the following text in the Results to emphasise the clinical potential of our findings (line 397):

“In summary, we have found a variety of candidate driver genes and pathways in BCAC, whereas the majority (94%) of BCA cases harboured one of 2 mutually exclusive hotspot mutations in genes involving the Wnt/ β -catenin signalling pathway. The presence of either mutation in BCA correlated with β -catenin IHC nuclear positivity. There is also a lower tumour mutation burden and few SCNAs in BCA compared to BCAC, in which we found recurrence of 16q and 5q copy number loss. These findings are summarised in Supplementary Figure 8. These distinctions, along with histopathology, may be useful in developing diagnostic biomarkers.”

We have also emphasised this in the last sentence of the Discussion section:

“Importantly, our work has highlighted that molecular profiling, along with histopathology and IHC, can be used for more accurate diagnosis of SGTs.”

After helpful feedback from the reviewers, we performed additional experiments and analyses and, therefore, have also created a mind map based on the functional experiments performed *in vitro* to characterise the *FBXW11* p.F517S and *CTNNB1* p.I35T mutations and studies of the tumours themselves with IHC staining and analysis of Wnt/ β -catenin gene targets using our RNA sequencing data (Supplementary Figure 12). It summarises the evidence showing that these recurrent mutations result in reduced binding of FBXW11 and β -catenin to each other and that β -catenin is not properly degraded and translocated to the nucleus where transcription is activated. We have referred to it in the last paragraph of the Results section.

“A schematic summary of the findings from our validation experiments, IHC studies and RNA sequencing analysis is shown in Supplementary Figure 12. Taken together, our findings strongly suggest that for 30/32 BCA cases (93.7%), activation of the Wnt/ β -catenin signalling pathway, either through mutation of *CTNNB1* at p.I35T or *FBXW11* at p.F517S (Figure 7), is the primary

mechanism of tumorigenesis. Notably, this molecular mechanism does not represent a major driver event in BCAC.”

Reviewer #2

Remarks to the Author:

The manuscript under review entitled “Activation of Wnt/-catenin Signalling by mutually exclusive FBXW11 and CTNNB1 hotspot mutations drives salivary gland basal cell adenoma” presents a comprehensive genetic profiling of two types of salivary gland tumors, BCA (benign) and BCAC (malignant carcinoma). Utilizing whole-exome and transcriptome sequencing, the study examines somatic variants, copy number alterations, germline variants, and gene fusions. Among the key findings are the identification of a novel mutational event involving elevated Wnt/-catenin Signalling and the observation of the FBXW11 p.F517S mutation, both of which are significant contributions to the field. However, the manuscript suffers from several shortcomings that impact its clarity.

Strengths:

1. Identification of Novel Mutational Events in BCA: The discovery of the FBXW11 mutation and its association with elevated /-catenin levels is a noteworthy finding. This adds valuable information to the understanding of genetic alterations in BCA salivary gland tumors.

2. Insight into Tumorigenesis in BCAC: The study's use of high-throughput sequencing technologies has generated high-quality genetic data, which is particularly important given the rarity of these tumor types. Moreover, the study documented the genetic changes in both the benign and malignant forms of these rare salivary gland tumors, providing new insights into the tumorigenic processes in these tumours.

General Comment:

The manuscript lacks a clear thematic focus. It frequently jumps between descriptions of gene mutations, associated pathways, and different cancer groups, making it difficult to follow. A more

structured approach, perhaps separating the description of genetic findings from mechanistic analyses, would enhance readability.

Based on the reviewer's suggestion, we have now separated the functional validation experiments from the genetic findings by describing the genetic findings in BCAC and the germline analysis prior to presenting the results of the functional analyses. We have also added the following sentence at the end of the "Overview of somatic alterations in BCA and BCAC" section in the results to guide the reader.

"The high-level comparison of BCA and BCAC described above alludes to important differences in these tumours. In the following subsections, we further describe and compare the driver genes and pathways in salivary gland BCA and BCAC, and present the results of the functional characterisation of the *CTNNB1* p.I35T (c.104T>C) mutation found in BCA and a novel recurrent *FBXW11* mutation identified in this study."

We have also added a brief summary (line 397) describing the recurrently mutated pathways and genes in BCAC, along with a supplementary figure summarising the molecular findings (as suggested by Review 1).

Finally, we have split the functional validation studies into two sections, with the first section describing the results of the MD simulations and the second section describing the *in vitro* analyses. We provide a sentence as a segue from the genetic analyses to these experiments (line 436).

"The *FBXW11* p.517S substitution is predicted to cause local perturbations that affect the β -catenin binding site

Given that the *FBXW11* p.F517S mutation has not been previously reported, we sought to functionally characterise this mutation along with the *CTNNB1* p.I35T mutation. ..."

Specific points:

1. While the investigation of β -catenin accumulation by FBXW11 pF517S is interesting, the study falls short by not conducting further experiments to demonstrate the effect of this mutation on β -catenin subcellular localization and its transactivation activity.

Following the reviewer's remark, we performed confocal laser scanning microscopy analyses and cell fractioning assays to provide further evidence of the increased β -catenin levels in cells overexpressing the FBXW11 mutant compared to cells overexpressing the wild-type FBXW11 (see Figure 5 and associated text). These experiments document the increased nuclear levels of β -catenin associated with the overexpression of mutant FBXW11. These experiments were repeated with the mutant CTNNB1, with similar results. The transactivation assays were hindered by technical challenges. However, after discussing this with the editor we were able to show, using the RNA sequencing data from tumours, that a number of Wnt/ β -catenin gene targets were upregulated in tumours with either the CTNNB1 p.I35T or FBXW11 p.517S mutation compared to tumours without these mutations. Notably, the tumours with the FBXW11 p.F517S mutation showed similar patterns of expression as those with the CTNNB1 p.I35T mutation. These results are shown in a new figure, Supplementary Figure 11. Importantly, the IHC staining data we have presented also correlated with the presence or absence of these mutations. IHC for β -catenin is commonly used to demonstrate aberrant expression of β -catenin, particularly in colorectal cancer. As described in the manuscript, tumours harbouring either of these mutations had little membranous staining for β -catenin and strong nuclear positivity, while membranous positivity and negative nuclear staining was observed in tumours without the mutations and in normal salivary gland cells (Figure 1 and Figure 6). Taken together, these data provide corollary evidence that the accumulation and nuclear localisation of β -catenin and transactivation of Wnt/ β -catenin gene targets is associated with the expression of the mutant CTNNB1 and FBXW11 genes in these tumours.

2. The mutually elusive nature of CTNNB1 at p.I35T and FBXW11 at p.F517S in BCA samples is observed but not further explored. The MD simulations showed that the CTNNB1 p.I35T substitution disrupts the hydrophobic interaction between amino acid residues I35 of CTNNB1 and A428 of FBXW11 (Figure 4a). Disruption of the interaction between CTNNB1 p.I35T and FBXW11 could be experimentally validated.

As recommended by the reviewer, co-IP assays were performed to validate the *in silico* observations. In line with the *in silico* predictions, we demonstrated a reduced amount of co-

immunoprecipitated β -catenin by FBXW11 with β -catenin^{I35T} compared to the WT protein (see Figure 5 and associated text). This finding mirrors the increased levels of the β -catenin in cells expressing β -catenin^{I35T} compared to the WT protein, as observed in cell fractioning experiments and confocal laser scanning microscopy (Figure 5e and Supplementary Figure 10).

3. The study presented the MD simulation analysis of FBXW11 at p.F517S and β -catenin. To demonstrate F517S affects the FBXW11 region involved in binding β -catenin, providing protein-ligand interaction stability throughout the simulation by RMSD (root mean square deviation) as well as RMSF (root mean square fluctuation) data of protein which depicts the flexibility for the wildtype and mutant protein when binding to β -catenin would greatly strengthen the conclusion. Additionally, providing the protein-ligand contacts throughout trajectory would improve clarity of the data.

In our analyses, following energy-minimization, WT FBXW11 and a model of the p.F517S FBXW11 mutant were subjected to 100 short (1 ns) independent MD simulations, each followed by energy re-minimization. Using this approach, no conspicuous conformational changes involving multiple residues could be detected, and RMSDs failed to show significant differences between the two modelled systems. However, there were local mutation-induced conformational changes, at the level of individual residues. Indeed, as correctly anticipated by the reviewer, plotting of the RMSF data provides evidence that Arg468 is the residue exhibiting the largest variation in mobility associated with the F517S substitution, in line with the panel originally showing the protein conformers. RMSF data analysis also allowed us to identify a second region with high flexibility differences (encompassing residues 263 and 264). This region maps within the first WD40 repeat, which is adjacent to F517 and includes Y265, a key residue that is directly involved in β -catenin binding.

Following the reviewer's advice, we revised the panels reporting on the structural analyses and molecular dynamics simulations to include a new panel (i.e., Figure 4b) which shows the RMSF data for each residue of the FBXW11 WD40 domain relative to the two modelled systems (WT vs mutant FBXW11) and the observed difference between them. Panel (a) was revised, accordingly.

Regarding the possibility to study also the effects on the binding of β -catenin, we deemed that the result showed an important alteration on a residue that concurs to the formation of the binding

site of β -catenin. Since the available structure of the β -catenin-FBXW11 complex includes only a small peptide of the ligand, which does not necessarily represent the entire intermolecular binding network stabilizing the complex, the latter was not included in the molecular dynamics simulations. It may be the subject of a future, more dedicated, study of the FBXW11/ β -catenin interaction.

4. To strength the conclusion that 30/32 BCA cases (93.7%) arose from accumulation and activation of the Wnt/ β -catenin signalling pathway, through mutation of CTNNB1 at p.I35T or FBXW11at p.F517S, the levels of β -catenin and FBXW11 in cancer tissues harboring the mutations of CTNNB1 at p.I35T or FBXW11 at p.F517S should be examined accompanying the IHC staining of β -catenin.

We reviewed the β -catenin IHC staining patterns for all the relevant tumours. In the main text, we have described the staining levels:

“All cases we tested that harboured a mutation in either gene appeared similar, all having little or no membranous positivity with variable mild to moderate cytoplasmic staining and variable strong nuclear positivity, particularly in peripheral palisading dark cells around tumour cell islands.”

In summary, while the manuscript presents significant genetic findings in the context of salivary gland tumors, its value is diminished by several issues listed. Future revisions should focus on enhancing the narrative structure, and incorporating functional experiments to elucidate the role of identified mutations in tumorigenesis.

Thank you for your comments. We believe that the additional functional experiments and structural analysis you suggested enhance the manuscript and provide not only a more complete picture of the roles of the recurrent *FBXW11* and *CTNNB1* mutations in BCA, but general insight into the mechanisms of dysregulation of the Wnt/ β -catenin signalling pathway, which plays an important role in a variety of tumour types.

Reviewer #3

Remarks to the Author:

Wong et al. present "Activation of Wnt/B-catenin signalling by mutually exclusive FBXW11 and CTNNB1 hotspot mutations drives salivary gland basal cell adenoma". It is a very nice study of the somatic genomic alterations in two extremely rare cancer entities of the salivary gland. Overall, the study is very well done, following-up on the most interesting new finding (the mutation in FBXW11) with computational and in vitro studies that support its potential oncogenic role.

I only have a few minor comments:

1 - There's a sentence to close one of the results sections that reads as follows:

"Taken together, we can conclude that 30/32 BCA cases (93.7%) arose from activation of the Wnt/ β -catenin signalling pathway, either through mutation of CTNNB1 at p.I35T or FBXW11 at p.F517S (Figure 1). Notably, this molecular mechanism does not represent a major driver event in BCAC."

I believe that this might be a bit too strong. While it is true that all the evidence suggests a very important role for these two mutations, there is no experimental demonstration that the BCA cases "arose from activation" of the Wnt pathway. I know that this is extremely hard to actually prove, but given the growing number of papers showing driver mutations in otherwise healthy cells, I believe that the expression "arose from" is not yet deserved. Maybe something like: "30/32 cases had alterations in the Wnt pathway, suggesting a very important role of this biological process in the tumorigenesis of BCA"?

We thank the reviewer for their advice. We have revised the text accordingly. In addition, based on suggestions from Reviewer 2, we have performed additional functional experiments, which lend support to the assertion that alterations in *FBXW11* and *CTNNB1* lead to the activation of the Wnt/ β -catenin signalling pathway. This pathway is a well-studied mechanism of tumorigenesis in many tumour types, including colorectal, endometrial and lung cancer (e.g. see PMID 35836256, 24309006, 32560059, 34068065) and IHC is frequently used to infer activation of the Wnt signalling pathway (e.g. see 20395444, 27574554). With the addition of the functional assays suggested by Reviewer 2, combined with IHC for β -catenin showing nuclear localisation of β -catenin in tumours, and evidence of elevated gene expression of Wnt/ β -catenin gene targets with our RNA sequencing data from tumours, we have now shown that both of the recurrent

FBXW11 and *CTNNB1* mutations in BCA have a downstream effect on the major steps of the Wnt/ β -catenin signalling pathway, including upregulation of Wnt/ β -catenin gene targets (Figure 1, Figure 5, Figure 7, Supplementary Figure 10 and Supplementary Figure 11).

It is important to note that the *in vitro* results we present concur with our observation that BCAs with either *CTNNB1* p.I35T or *FBXW11* p.F17S are positive for nuclear β -catenin staining, indicating that in these tumours, β -catenin has been translocated to the nucleus in these tumours. In contrast, in tumours without either mutation, β -catenin was localised to the membrane, indicating that Wnt/ β -catenin signalling was not activated. The IHC data also concurred with our RNA sequencing data, which showed that targets of the Wnt/ β -catenin signalling pathway were expressed at higher levels in tumours with either mutation. Specifically, we have shown (1) the *FBXW11*^{F517S} mutant protein has reduced binding to β -catenin, (2) the *CTNNB1*^{I35T} (β -catenin^{I35T}) mutant protein has reduced binding to *FBXW11*, (3) cells expressing the mutant proteins have reduced polyubiquitination of β -catenin, (4) β -catenin accumulates at higher levels in cells expressing either mutant protein (5) higher levels nuclear β -catenin are found in cells expressing either mutant protein and in tumours with either mutation, and (6) in tumours, an increased level of β -catenin is associated with transactivation of Wnt/ β -catenin gene targets. These findings have been summarised in a 'mind map', as suggested by Reviewer 1 (Supplementary Figure 12).

In addition, the BCA tumours had low tumour mutation burden, few fusion genes and few somatic copy number alterations (Figure 1, Figure 2 and Supplementary Figure 4). Other than *CTNNB1* and *FBXW11* mutations, found in 94% of the BCAs, *FGGY* and *CDKN1C* were found (by oncodriveFML only; see #2 below) to be significantly mutated, but were mutated in only 2/32 samples each (Supplementary Table 5). This, together with our functional validations, make it highly unlikely that there are other major drivers of tumourigenesis in the BCA cases harbouring the recurrent *CTNNB1* or *FBXW11* mutations, which constitute ~94% of BCA cases in our cohort. Considering the reviewer's comment we have altered the text as follows:

"A schematic summary of the findings from our validation experiments, IHC studies and RNA sequencing analysis is shown in Supplementary Figure 12. Taken together, our findings strongly suggest that for 30/32 BCA cases (93.7%), activation of the Wnt/ β -catenin signalling pathway, either through mutation of *CTNNB1* at p.I35T or *FBXW11* at p.F517S, is the primary mechanism of tumourigenesis in this tumour entity. Notably, this molecular mechanism does not represent a major driver event in BCAC."

2 - There's a variety of tools to identify cancer driver genes. Why did the authors only use dn/ds? This can lead to missing important genes.

Based on the reviewer's comment, we have run additional analysis with the latest version of oncodriveFML (see Methods). The results are provided in Supplementary Table 5b. In the BCAC cohort, oncodriveFML identified *CYLD*, *PIK3CA* and *IKBKB* as significantly mutated ($q < 0.1$). While we had already discussed in the original manuscript the potentially important role these genes play in BCAC, this additional analysis strengthens our argument. In the BCA cohort, *CTNNB1* and *FBXW11* were identified by oncodriveFML as candidate drivers, as previously found using dNdScv, while *CDKN1C* and *FGGY* both had q -values of 0.087, but were mutated in only 2/32 BCA cases each. We have made the appropriate changes to the Results and Discussion in the manuscript. We note that sequencing of additional cases of these rare tumours will provide more statistical power to identify additional drivers, particularly for BCAC.

Figure 1 was also modified to add *RPL22* (previously identified by dNdScv) as it is also a COSMIC CGC gene but inadvertently excluded from Figure 1 (*MED13L* and *PCDH9* are not CGC genes and not shown in Figure 1; shown in Supplementary Table 5b).

3 - There is a sentence in the discussion: In this study, CYLD mutations were found in only 2 of 11 BCAs and none of the 32 BCACs. I believe that here BCA / BCAC is switched

Thank you for spotting this on line 484/485 in the original manuscript. We have made the correction to the line.

We thank Reviewer 3 for their very constructive comments.